# Understanding Scale Shift in Domain Generalization for Crowd Localization

## Abstract

Crowd localization plays a crucial role in visual scene understanding towards predicting each pedestrian location in a crowd, thus being applicable to various downstream tasks. However, existing approaches suffer from significant performance degradation due to differences in head scale distributions (scale shift) between training and testing data, a challenge known as domain generalization (DG). This paper aims to comprehend the nature of scale shift within the context of domain generalization for crowd localization models. To this end, we address three key questions: (i) how to quantify the scale shift influence on DG task, (ii) why does this influence occur, (iii) how to mitigate the influence. Specifically, we first establish a benchmark, ScaleBench, and reproduce 20 advanced DG algorithms, to quantify the influence. Through extensive experiments, we demonstrate the limitations of existing algorithms and highlight the under-explored nature of this issue. To further understand its behind reason, we provide a rigorous theoretical analysis on scale shift. Building on this analysis, we further propose a simple yet effective algorithm called Semantic Hook to mitigate the influence of scale shift on DG, which also serves as a case study revealing three significant insights for future research. Our results emphasize the importance of this novel and applicable research direction, which we term *Scale Shift Domain Generalization*.

## 1 Introduction

Crowd localization (Liu et al., 2019; Gao et al., 2020; Song et al., 2021; Liang et al., 2022; Han et al., 2023; Chen et al., 2024) aims to accurately identify the positions of individuals, particularly in dense and diverse population scenarios. It provides quantity of applicable utilities for downstream tasks. For example, pinpointing the exact location of each individual within a crowd can improve public surveillance (Li et al., 2013), facilitate event management (Mundhenk et al., 2016), and assist in urban planning (Marsden et al., 2018). Moreover, the frameworks for crowd localization are applicable to dense cell (Morelli et al., 2021) and pathology detection (Lagogiannis et al., 2023), thereby advancing clinical diagnosis. Hence, previous researchers have developed a variety of supervised crowd localization algorithms.

However, the generalization performance of these fully-supervised models often fall short when exposed to unseen data distributions, a challenge commonly referred to as *domain shift* (Wang et al., 2022). Over the years, the community has made substantial efforts to address various forms of domain shifts, such as dataset shifts (Du et al., 2023) (e.g., from SHHA (Zhang et al., 2016) to QNRF (Idrees et al., 2018)), scene shifts (Wang et al., 2019; Gong et al., 2022) (e.g., from street to stadium), and weather shifts (Peng & Chan, 2024) (e.g., from sunny to snowy). It is widely accepted that such domain shifts between the training (source) and testing (target) domains can lead to performance degradation in crowd analysis models. Recently, Ma et al. (2021) have identified that the *head scale* distribution of crowd datasets significantly influences the performance of crowd analysis models when crossing datasets evaluation. However, it is still unexplored how scale shift affects the performance under **domain generalization** (Wang et al., 2022) scenario.

Hence, we conduct realistic experiments to reveal that the generalization performance of state-of-the-art (*sota*) crowd locators degrade significantly when scale shift occurs across domain. Specifically, as shown by Table 1, we firstly divide existing datasets into two domains (like *Tiny* and *Big*)

Table 1: Localization $F_1$ score (%) results in the scale shift scenario, where $A \mapsto B$ indicates that the model is trained and validated on domain $A$ and tested on domain $B$. When $A = B$, this denotes the in-distribution (InD) situation; otherwise, it indicates out-of-distribution (OOD). The *Tiny* and *Big* represents the two domains, with head scale distribution difference. The values in the brackets denote the performance degradation from InD to OOD. See Appendix D.5 for detailed setting.

| Setting | Scale Distribution KL-Divergence | IIM (Gao et al., 2020) | P2PNet (Song et al., 2021) | CLTR (Liang et al., 2022) | SteererNet (Han et al., 2023) | PET (Liu et al., 2023) |
|---|---|---|---|---|---|---|
| Tiny $\mapsto$ Tiny | 0.02 | 62.05 | 58.15 | 70.90 | 78.52 | 62.32 |
| Big $\mapsto$ Tiny | 18.36 | 11.25 (50.80↓) | 12.00 (46.15↓) | 9.71 (61.19↓) | 47.59 (30.93↓) | 10.42 (51.90↓) |
| Big $\mapsto$ Big | 0.45 | 83.46 | 73.17 | 80.77 | 93.27 | 79.96 |
| Tiny $\mapsto$ Big | 17.35 | 62.20 (21.26↓) | 41.72 (31.45↓) | 49.12 (31.65↓) | 69.52 (23.75↓) | 43.87 (36.09↓) |

according to their average head scale[1]. And we independently train two crowd locators on the training set of *Tiny* and *Big* domains, and we test their corresponding localization performance on the test set of *Tiny* or *Big* domains. Take the performance on the *Tiny* domain' s test set as an example. When PET (Liu et al., 2023) is trained on *Tiny* domain' s train set, its $F_1$ score is $62.32\%$, while this metric decreases to $10.42\%$ when training set is from *Big* domain, with a performance degradation of $\mathbf{51.9\%}$. And we observe consistency phenomenon over other *sota* locators, which strongly support the significance of scale shift for domain generalization.

Despite recognizing the impact of scale shifts on domain generalization, this issue has not been sufficiently addressed in the literature. Previous work mainly concentrate on how to capture different scales in a fully-supervised paradigm (Han et al., 2023; Wang et al., 2023a). As for the scale shift under cross dataset evaluation, SDNet (Ma et al., 2021), it focuses on "domain adaptation"[2], in which the target domain is accessible during training. Our task "domain generalization" assumes the whole target domain should be *unseen* during training, which is more pertinent to the deployment of crowd models in open-set environments. Furthermore, much of the existing research on cross-domain crowd analysis (Du et al., 2023; Gong et al., 2022; Du et al., 2023; Peng & Chan, 2024) overlooks a crucial aspect: there is no assurance of performance retention on the source domain. In real applications where crowd locators may operate in open-set scenarios, the target scales remain uncertain. Thus, it is essential to maintain performance on both out-of-distribution and in-distribution data. Therefore, it is critical to answer: **How can we effectively generalize crowd localization models to unseen scales while preserving performance on seen scales?**

In this paper, we present as far as we know the FIRST study on scale shift domain generalization in crowd localization. Our research addresses three key questions: 1) *Influence:* How to quantify the influence of scale shift on the domain generalization performance of crowd localization? 2) *Analysis:* Why does this influence occur? 3) *Mitigation:* What strategies can be employed to mitigate this influence? We provide a comprehensive analysis to answer these questions.

- **Influence: ScaleBench as a benchmark to quantify scale shift and its influence.** In Sec. 2, we establish a scale benchmark dataset "ScaleBench" with 17,138 images to officially quantify scale shift and its influence on domain generalization with crowd localization tasks. Specifically, we manually annotate over 1.5 million bounding boxes for datasets (SHHA (Zhang et al., 2016), SHHB (Zhang et al., 2016), and QNRF (Idrees et al., 2018)) and integrate with originally annotated datasets (SHRGBD (Lian et al., 2019), JHU (Sindagi et al., 2020), and NWPU (Wang et al., 2020b)). Furthermore, we propose an innovative domain partitioning method to categorize the images in ScaleBench into four distinct domains based on progressive scale distributions. This benchmark is then utilized to evaluate domain generalization ability under scale shift conditions. Then, we designed a PyTorch codebase and conducted a comparative experiments of 20 state-of-the-art domain generalization algorithms[3], most of which exhibits even worse performance than baseline, thus reveals the under-studied nature of this issue.

---

[1] See Sec. 2 for details.

[2] SDNet includes domain adaptation and test-time domain adaptation. See Sec.3.1 and 3.2 of Wang et al. (2022) for detailed task difference with domain generalization.

[3] Codebase has been attached to the supplementary material, and will be open-sourced along with annotated dataset after double-blind review.

- **Analysis: Scale Shift as Mixed Shifts in Diversity and Correlation.** In Sec. 3.1, we investigate the reasons behind the unsatisfactory performance of domain generalization models and find that scale shift affects domain generalization by causing the model to learn a spurious association between *scale* and *target*. By employing established definitions of domain shifts, which include diversity and correlation shifts (Ye et al., 2022), we prove that scale shift embodies a combination of both. This elucidates why existing domain generalization algorithms struggle with scale shifts.
- **Mitigation: Semantic Hook as a novel solution and case study for future works.** In Sec. 3.2, we introduce an algorithm, Semantic Hook, designed to strengthen the association between semantic features and task predictions. Using Semantic Hook as a case study, our extensive analysis provides three key insights for future research on scale shift domain generalization: *1) Enhancing the connection between final predictions and semantic features while minimizing scale feature influence. 2) Traditional image interpolation methods, while useful, have limited efficacy. 3) Increasing training data yields marginal benefits if the data maintain a consistent scale distribution.*

## 2 SCALE BENCH

### 2.1 PROBLEM FORMULATION OF SCALE SHIFT DOMAIN GENERALIZATION

Under the domain generalization scenario, given the source $\mathcal{D}_{src}$ and target $\mathcal{D}_{tra}$ domains, we acknowledge that the object scale distributions differ between the source and target domains: $p_{src}(c|z) \neq p_{tar}(c|z)$, where $z$ denotes the object and $c$ represents the object scale. For instance, the head scales in the source domain may be smaller compared to those in the target domain. With this setting, defining domain distribution $\mathcal{D}_{src/tra}$ as the joint distribution of input $\mathcal{X}$ and target $\mathcal{Y}$, domain generalization necessaries to train a model $h : \mathcal{X} \longmapsto \mathcal{Y}$ on source domain $\mathcal{D}_{src}$, which will perform well on target domain $\mathcal{D}_{tar}$. In this paper, however, we go beyond the standard domain generalization setting and aim for the learned model $h$ to simultaneously maintain performance in both the source domain and the target domain, which consist of data with diverse scale distributions. Formally, we formulate this as a constrained optimization problem:

$$h^* = \arg\min_{h \in \mathcal{H}} \mathbb{E}_{(x_s, y_s) \sim \mathcal{D}_{src}} \mathcal{L}(h(x_s), y_s), \quad \text{s.t.} \quad \mathbb{E}_{(x_t, y_t) \sim \mathcal{D}_{tar}} \mathcal{L}(h(x_t), y_t) < r_{ood}, \quad (1)$$

where $r_{ood}$ denotes the upper bound of out-of-distribution (OOD) generalization risk.

### 2.2 SCALE DETERMINED DOMAIN PARTITION

Although scale shift is critical for generalization and numerous crowd datasets have been released, there is no existing dataset specifically related to scale shift that meets the strict requirements of our setting (see Sec. 2.1) for an in-depth study of scale variance. Therefore, we establish the FIRST dataset benchmark, specifically designed to address the scale shift problem by overcoming the following two non-trivial challenges. Figure 1 illustrates the pipeline in building the benchmark.

#### 2.2.1 CHALLENGE 1: ABSENCE OF SCALE ANNOTATION IN MAINSTREAM DATASETS

To study the scale shift problem, it is crucial to have bounding box information for each human head, as these bounding boxes include size information essential for exploring scale variance. However, we cannot always obtain such information from temporal mainstream datasets. Earlier but still widely-used datasets such as SHHA (Zhang et al., 2016), SHHB (Zhang et al., 2016), and QNRF (Idrees et al., 2018) primarily involve manual annotations marking a single point at the center of each head. Fortunately, more recent datasets, including SHRGBD (Lian et al., 2019), JHU (Sindagi et al., 2020), and NWPU (Wang et al., 2020b), provide bounding boxes for each head. Nevertheless, despite the large number of images included in these more detailed datasets, they may still fail to cover the full range of scale variations due to the inherent smoothness and continuity of the scale attribute.

To address this limitation, we have conducted manual annotations for SHHA, SHHB, and QNRF, adding bounding boxes to supplement our understanding of object scales. For further details regarding the annotation process, please refer to the Appendix H.1. In total, we have provided bounding box annotations for **1.5 million objects across 2,700 images**. By combining these newly annotated datasets with three existing datasets, we create a rich data resource with 17,138 images that forms the foundation for our ScaleBench.

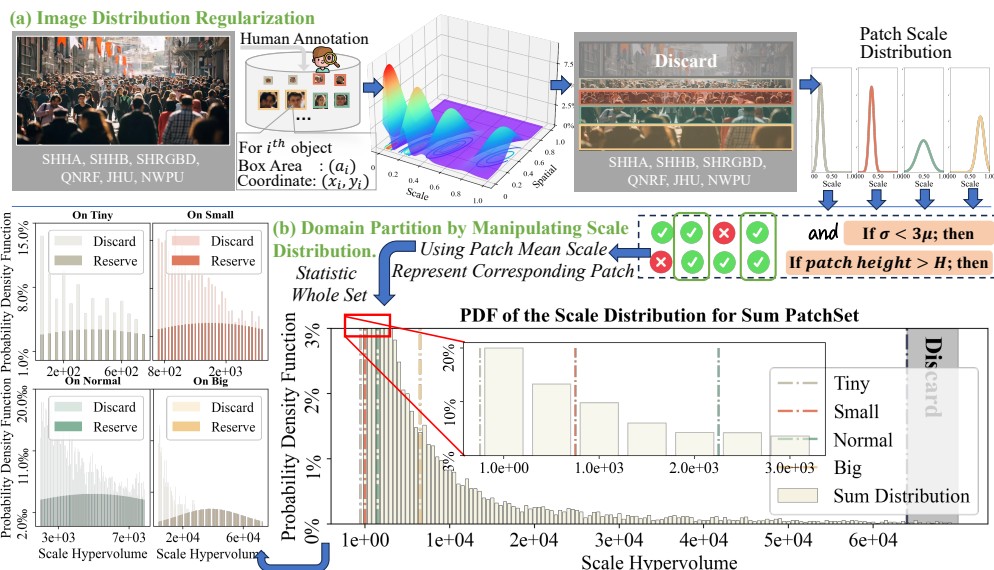

Figure 1: Pipeline for generating domains for ScaleBench, along with the scale statistics of ScaleBench. First, we regularize the image-level scale distribution as shown in (a) and filter out unqualified samples. We then analyze the overall scale distribution, which is subsequently divided into four distinct domains with inter-domain scale shifts, as illustrated in (b).

### 2.2.2 CHALLENGE 2: CONTINUAL SCALE DISTRIBUTION PARTITION

While we currently have access to a rich repository of data, the next challenge lies in effectively partitioning this data into domains that accurately reflect scale shifts. The simplest approach is to collate objects within a dataset, derive a scale distribution, and then apply various scale thresholds for partitioning. However, this paradigm faces two significant challenges: 1) the presence of varying scale ranges within a single image; and 2) the complexities involved in selecting appropriate scale thresholds. Specifically, the original images in existing datasets often exhibit intrinsic scale variation, meaning that each image encompasses objects spanning multiple scale levels. Consequently, assigning such images to different domains does not guarantee sufficient scale differentiation (Wang et al., 2022) among them. Moreover, the choice of scale thresholds directly impacts the number of samples in each domain, with improper selection leading to imbalances in domain representation.

To address these challenges, we propose a novel controllable domain scale partitioning module, along with an evaluation pipeline designed to leverage it. We shall introduce these in details below.

**Controllable Domain Scale Partition**  To achieve scale controllable domain partition, we first propose an Image-Level Scale Distribution Regularization, that aims to eliminate intra-image scale variance by dividing an image into intra-scale consistent patches. Then, we set these patches as our new *images*, and propose a Domain Partition by Manipulating Scale Distribution to group those patches into several domains. Let us elaborate on the processes within our proposed framework.

*1) Image-Level Scale Distribution Regularization.* As aforementioned, the significant scale variance present within individual images complicates the assignment of these images to scale-aware domains. We attribute this challenge to the high resolutions of original images collected from prior research; for example, some images in the NWPU dataset are more than $10,000^2$ pixels. To mitigate this, we propose segmenting images into patches according to scales. This reduces the extensive image-level scale variation and enables better regularization of the sample-wise scale distribution. Importantly, this patch division does not affect the subsequent training process, as temporal locators (Liang et al., 2022; Han et al., 2023) operate by cropping images into patches for training.

To this end, we utilize a mixed Gaussian model (Reynolds et al., 2009) to approximate the image-level scale distribution $p(c)$ following (Wang et al., 2023a):

$$p(c) = \sum_{k=1}^{K} \omega_k \cdot \mathcal{N}(c_k|\mu_k, \sigma_k), \text{where} \sum_{k=1}^{K} \omega_k = 1, \quad (2)$$

in which $K$ is a pre-defined number of sub-Gaussian distribution $\mathcal{N}$, and $\omega_k$ denotes the learned weight over $k$ sub-distribution. With this Eq. 2, we can derive $K$ scale distributions $\{p_k(c)|p_k(c) \sim \mathcal{N}(\mu_k, \sigma_k)\}_{k=1}^K$, where each individual one could be recognized as a Gaussian distribution. However, solely employing a one-dimensional mixed Gaussian model risks losing spatial information about the objects, leading to sub-Gaussians that lack spatial compactness, complicating the identification of each sub-Gaussian when splitting images. Thus, we opt for using a two-dimensional mixed Gaussian model to fit the joint distribution over scale $p(c)$ and spatial location $p(l)$ simultaneously.

$$p(c,l) = \sum_{k=1}^K \phi_k \cdot \mathcal{N}(c_k, l_k|\vec{\mu}_k, \Sigma_k), \text{where} \sum_{k=1}^K \phi_k = 1, \tag{3}$$

Following (Wang et al., 2023a), the spatial location distribution $p(l)$ focuses on the vertical coordinates of the objects.[4] With this approach, we can derive the $K$ instances of sub-joint distributions $\{p_k(c,l)\}_{k=1}^K$, incorporating both scale and spatial information.

To partition the original images into $K$ patches, we proceed using the boundaries of the sub-spatial distributions $(\min l_k, \max l_k)$. After splitting the images, we apply two filtering criteria to eliminate unqualified patches: first, we discard any patch where the intra-patch scale distribution has a standard deviation exceeding three times the mean (3-$\sigma$ criteria). Second, patches with minimal height are also filtered out to ensure they are suitable for training locators. **This process reduces scale variance within each individual patch, allowing us to represent the entire patch by its mean scale in generating scale distribution of whole dataset** (see main distribution in Figure 1).

*2) Domain Partition by Manipulating Scale Distribution.* Then, with regularized patches as our new *images*, we proceed to separate them into domains. Our framework commences from patch-set scale distribution, which is derived by counting the frequency of patch mean scales, with $f(c)$ as its Probability Density Function (PDF). To obtain $M$ domains to support the study of the domain generalization, we split $f(c)$ equally over the sample number, in case of sample imbalance among domains. And the PDF of scale distribution in $m^{th}$ domain can be presented by:

$$f_m(c) = f(c), \text{where } c \in [c_{m-1}, c_m], \int_{c_{m-1}}^c f(c)\mathrm{d}c = \frac{1}{M}. \tag{4}$$

By splitting as Eq. 4, we can derive $M$ domains with intra-domain compact scales. However, to further study the domain generalization issue, enhancing the distance between any two domains facilitates the alignment with the theoretical definition (Wang et al., 2022) to the issue. To that effect, we further conduct a Gaussian sampling on each $f_m(c)$ as:

$$f'_m(c) = \mathcal{G}_1\left(\frac{c_m + c_{m-1}}{2}, \sigma_m\right) \odot f_m(c), \text{where } \sigma_m = \arg\max_\sigma \int_{c_{m-1}}^{c_m} f'_m(c)\mathrm{d}c,$$

$$\text{s.t.,} \forall m_1, m_2 \leq M, |\int_{c_{m_1-1}}^{c_{m_1}} f'_{m_1}(c)\mathrm{d}c - \int_{c_{m_2-1}}^{c_{m_2}} f'_{m_2}(c)\mathrm{d}c| \leq \epsilon, \tag{5}$$

in which the $\mathcal{G}_1$ denotes an one-dimensional Gaussian kernel, $\odot$ denotes the dot product, and $\epsilon$ is a very small error value. Empirically, we achieve $M$ instances optimal variance $\sigma_m$ by heuristic search. By now, we derive $M$ instances domains with $f'_m(c)$ as the PDFs of their scale distribution, which is adopted as the dataset to support the ScaleBench.

**ScaleBench Evaluation** To better validate the scale shift domain generalization over different scales, we divide the whole set into $M = 4$ domains. By this way, each domain is iteratively isolated as the target domain, and the remaining three are merged as a training domain, ensuring that the final results remain scale-unbiased.

During each iteration, the training process begins with the three source domains, which are further split into training and validation sets. After completing the training, the best-performing model is chosen based on its performance on the joint validation sets, representing its performance on in-distribution (InD) scales. Subsequently, we assess the performance of the selected model on the entire target domain, treating this result as its performance on out-of-distribution (OOD) scales. By

---

[4]This part will be further discussed in Appendix E.2.

averaging the InD and OOD performance across all iterations where each of the four domains serves as the target domain, we arrive at a final evaluation of generalization performance.

With above complete ScaleBench, we further developed a standard PyTorch-based codebase tailored for scale shift domain generalization tasks. Additionally, we have reproduced 20 state-of-the-art domain generalization algorithms and integrated them with a robust crowd localization baseline (Gao et al., 2020). Empirical results exhibited in Table 2 reveal a noteworthy trend: many domain generalization methods perform even worse than the baseline algorithm, highlighting the under-explored nature of the scale shift domain generalization issue. While we could not reproduce every algorithm, we welcome contributions to enhance our algorithmic repository.

## 3 HOW FAR DID WE GO ON SCALE SHIFT: DEFINITION AND A SOLUTION

The unsatisfactory performance of extensive domain generalization algorithms compels us to deepen our understanding of the domain scale shift issue. To that end, we provide theoretical analysis that connect scale shifts with classic domain shifts in Sec. 3.1. Furthermore, we propose a straightforward framework in Sec. 3.2, aimed at mitigating the negative influence of scale shift on domain generalization task, and offering guidance for future research in this area.

### 3.1 DOMAIN SCALE SHIFT

To better understand domain scale shift, we first need to answer:*Why does domain scale shift affects the generalization of crowd locators?* Crowd images are composed of numerous independent individuals $z$, each defined by various attributes, including semantic features $s$ (such as skin color), scale $c$, and other characteristics. Therefore, when we feed training sample pairs $(x, y)$ into a crowd locator, our learning process is modeled as the conditional distribution $p(y|x)$. Additionally, the input distribution $p(x)$ can be decomposed into a joint distribution of these various attributes, represented as $p(s, c, \dots)$. According to the chain rule, we can express this relationship as follows:

$$p(y|x) = \int_z p(y|z) = \int_{s,c,...} p(y|s,c,...) = \int_{s,c,...} \frac{p(y,s,c,...)}{p(s,c,...)} = \int_{s,c,...} \frac{p(c|y,s,...)p(y,s,...)}{p(s,c,...)}.$$
(6)

Let us elaborate on the derived term. The components $p(c|y, s, ...)$ and $p(s, c, ...)$ are related to scale $c$ and influencing the modeling of $p(y|x)$. Consequently, when domain shift occurs, variations in either $p(c|y, s, ...)$ or $p(s, c, ...)$ can lead to $p_1(y|x) \neq p_2(y|x)$. For the first term, $p(c|y, s, ...)$ stands for the same objects differing only in scale. We assert that this may not pose a significant problem, as it can often be mitigated through image interpolation. [5] However, the second term $p(s, c, ...)$, can cause the model to learn a spurious association between the output $y$ with scale $c$. As a result, when a scale shift occurs across domains, the learned spurious association $c \mapsto y$ in the source domain may fail to generalize to the target domain, leading to performance degradation.

According to the out-of-distribution (OOD) community, these spurious associations can lead to two types of domain shifts: diversity shift and correlation shift (Ye et al., 2022). In this context, Theorem 1 shows the spurious association of $c \mapsto y$ results in both two shifts simultaneously, whose detailed proof can be found in Appendix A.

**Theorem 1 (Scale Shift as A Mixed Domain Shift)** *For any two crowd domains, when scale distribution $p_1(c|z) \neq p_2(c|z)$, we have:*

$$Div_{div}(p_1, p_2) = \frac{1}{2} \int_{\mathbb{R}^c} |p_1(c) - p_2(c)| > 0 \qquad \text{(Existence of Diversity Shift)}$$

$$Div_{cor}(p_1, p_2) = \frac{1}{2} \int_{\mathbb{R}^c} \sqrt{p_1(c) \cdot p_2(c)} \sum_{y \in \mathcal{Y}} |p_1(y|c) - p_2(y|c)| \mathrm{d}c > 0,$$

$$\text{(Existence of Correlation Shift)}$$
(7)

*where Div denotes the divergence between two distributions.*

---

[5]This analysis highlights why image interpolation offers only limited help in our task.

Hence, the joint shift incurred from the spurious association of $c \mapsto y$ leads to the poor domain generalization performance over scale shift.

## 3.2 Semantic Hook: Enhancing Semantic Feature Association with Target

Based on Theorem 1, the coexistence of diversity and correlation shifts complicates the alignment of scale shift. Rather than alleviating spurious association between $c \mapsto y$, we focus on enhancing the semantic association $s \mapsto y$ to facilitate the learning of a scale-generalized model.

Our proposed method consists of two main components: a baseline learning loss that ensures standard performance on the source domain, and a semantic feature hook designed to extract and strengthen semantic features from image embeddings to improve the final prediction.

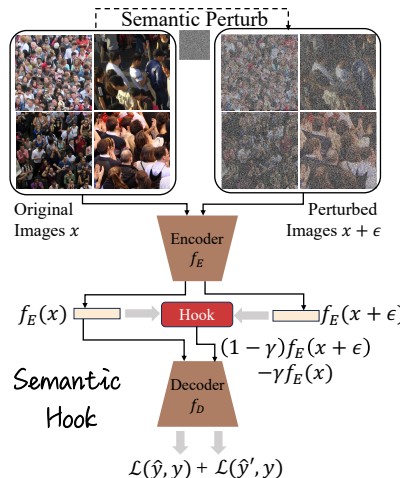

Specifically, given an image $x$, we utilize a standard encoder-decoder architecture typical of crowd locators to generate a prediction $\widehat{y} = f_D(f_E(x))$. The initial objective is to minimize the standard loss $\mathcal{L}(\widehat{y}, y)$. However, solely minimizing this loss can blur the learned image embedding, which contains both semantic and scale-related features. To enhance the semantic association, we extract semantic features from the image embedding while minimizing the impact of scale-related information. We achieve this by applying domain-shared Gaussian noise $\epsilon \sim \mathcal{N}(\lambda, \mathbf{I})$ on input $x$ to perturb it, resulting in a new embedding $f_E(x + \epsilon)$. Next, we define a modified prediction $\widehat{y}'$ as:

$$\widehat{y}' = f_D[(1 - \gamma)(f_E(x + \epsilon) - \gamma f_E(x))], \quad (8)$$

Figure 2: Training pipeline of our proposed SemanticHook.

where $\gamma$ is a coefficient that adjusts the weight of semantic features. With this new prediction, based on above standard loss, we further reduce the $\mathcal{L}(\widehat{y}', y)$.

**Intuitive Remark:** The added perturbation $\epsilon$ affects only the pixel values, which primarily influences the semantic information of the original image. Thus, the term $f_E(x + \epsilon) - \gamma f_E(x)$ represents the variation in the semantic representation due to the perturbation. This residual embedding tends to contain less task-specific information. However, by boosting the association of the predictions obtained from this residual embedding with the ground truth, we can potentially *hook* the task-relevant features from $f_E(x + \epsilon)$ to reduce the loss. Given that $\epsilon$ predominantly influences semantic information, the hooked task-relevant features are likely to be drawn from the semantic representation.

## 4 Experiment

### 4.1 Preliminary on Experiment

**Dataset** As aforementioned, we gather all of the samples from SHHA, SHHB, SHRGBD, QNRF, JHU, and NWPU. And we utilize the proposed pipeline to generate ScaleBench, where four scale differing domains are included. According to the average scale of each domain, we name the four domains as *Tiny* (T), *Small* (S), *Normal* (N), and *Big* (B), see Figure 1 for real scale distribution.

**Experimental Setting** By utilizing these four domains, we can evaluate performance by iteratively designating each domain as the target while treating the remaining three as the source domain (Leave-One-Out setting). Following DomainBed (Gulrajani & Lopez-Paz, 2021), we further split the training and validation set within each domain. When one domain is selected as the target domain (test set), its whole set will be utilized as testing samples. As for the baseline crowd localization method, we utilize a simplified paradigm[6] proposed in IIM (Gao et al., 2020), which is also widely adopted in (Gao et al., 2022; Wang et al., 2023a; Gao et al., 2023; Zhang et al., 2023a; Wen et al., 2024). And the detailed experimental setting is reported in the Appendix D.

---

[6]This will be discussed in Appendix B.

Table 2: $F_1$ score results on ScaleBench using the HRNetW-48 backbone model. The settings follow a Leave-One-Out experimental approach (Gulrajani & Lopez-Paz, 2021), where each model is trained on the training set of source domains, and tested on the target domain. Such as in the *Tiny* column, the InD performance are the results on the test set of source domains (*Small*, *Normal*, and *Big*), while the OOD performance indicates the results on the whole set of *Tiny* domain. The best results among algorithms are highlighted in **bold**, while the second-best results are underlined.

| Algorithm | Tiny | | Small | | Normal | | Big | | Global Avg |
|---|---|---|---|---|---|---|---|---|---|
| | InD | OOD | InD | OOD | InD | OOD | InD | OOD | |
| ERM | 87.32 | 58.05 | 75.08 | 85.30 | 77.87 | 87.90 | **79.26** | 81.57 | 79.04 |
| CORAL | 86.63 | 57.88 | 72.45 | 84.46 | 75.04 | 87.37 | 76.67 | 82.12 | 77.83 |
| DANN | 77.24 | 39.18 | 56.74 | 74.79 | 61.20 | 81.05 | 61.29 | 73.14 | 65.58 |
| MMD | 69.39 | 33.47 | 55.79 | 72.70 | 58.87 | 74.37 | 60.27 | 57.01 | 60.23 |
| IRM | 87.23 | 57.65 | 75.15 | 85.20 | 77.77 | 87.85 | 78.90 | 81.38 | 78.89 |
| SagNet | 86.80 | 57.70 | 73.97 | 85.30 | 76.66 | 87.49 | 77.59 | 79.03 | 78.07 |
| VREx | 87.14 | 58.77 | 75.07 | 85.24 | 76.91 | 87.63 | 78.82 | **82.56** | 79.02 |
| Mixup-F | 39.27 | 8.65 | 28.23 | 27.74 | 33.23 | 45.31 | 33.42 | 19.53 | 29.42 |
| Mixup-I | 86.18 | 56.05 | 72.78 | 84.64 | 75.38 | 87.71 | 77.36 | 78.69 | 77.35 |
| SAM | 86.77 | 57.36 | 73.14 | 85.75 | 75.63 | 87.96 | 77.43 | 75.51 | 77.44 |
| EFDM-I | 86.83 | 56.43 | 71.92 | 85.04 | 74.97 | 87.60 | 76.13 | 79.69 | 77.33 |
| EFDM-F | 86.97 | 56.55 | 71.78 | 85.13 | 75.48 | 87.44 | 76.19 | 80.22 | 77.47 |
| InfoBot-E | 85.91 | 55.56 | 72.04 | 84.72 | 75.32 | 87.20 | 76.37 | 78.03 | 76.89 |
| InfoBot-I | 85.92 | 55.50 | 71.43 | 84.52 | 74.60 | 87.02 | 75.82 | 77.54 | 76.54 |
| GAM | 85.22 | 50.36 | 68.77 | 84.55 | 72.43 | 86.96 | 73.21 | 69.81 | 73.91 |
| SAGM | **87.72** | 55.15 | 72.66 | 85.95 | 74.74 | 87.91 | 76.36 | 70.57 | 76.38 |
| CausalIRL-M | 76.67 | 41.16 | 59.30 | 77.24 | 63.00 | 79.24 | 65.91 | 67.81 | 66.29 |
| CausalIRL-G | 76.13 | 40.67 | 60.72 | 77.76 | 63.91 | 79.97 | 64.71 | 66.81 | 66.33 |
| SD | 86.12 | 55.40 | 72.62 | 84.21 | 75.56 | 87.22 | 76.87 | 79.98 | 77.25 |
| DomainDrop | 83.10 | 45.97 | 65.67 | 82.46 | 68.95 | 84.81 | 69.82 | 76.35 | 72.14 |
| SemanticHook (Ours) | 87.63 | **59.26** | **75.69** | **85.90** | **78.68** | **88.03** | 78.94 | 81.19 | **79.41** |

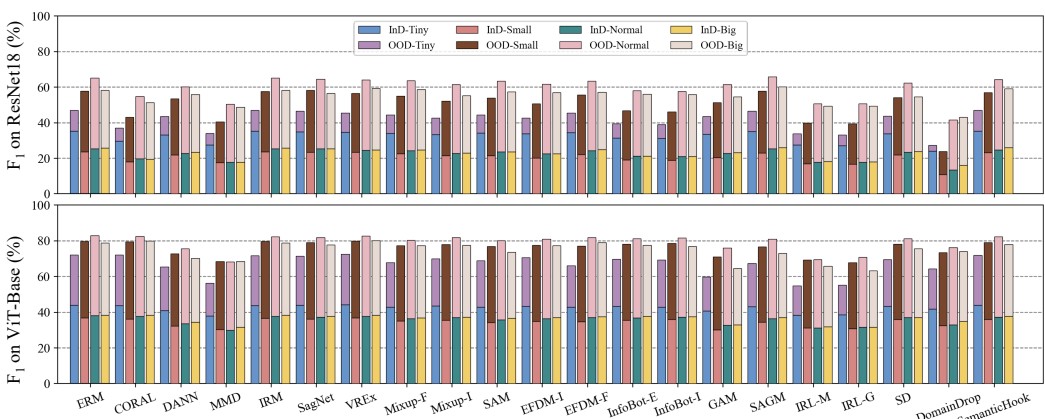

Figure 3: This stacked bar chart illustrates the $F_1$ scores for both InD and OOD results on ScaleBench. The height of each bar represents the average $F_1$ score across InD and OOD. To differentiate the contributions of each component, we use distinct colors for each section of the bars.

## 4.2 MAIN RESULTS

In Table 2 and Figure 3, we reproduce 20 out-of-distribution (OOD) algorithms on ScaleBench using three backbones: ResNet18 (He et al., 2016), HRNetW-48 (Wang et al., 2020a), and ViT-Base (Dosovitskiy et al., 2021). Given that the original architectures of ResNet18 and ViT-Base are not designed for dense prediction tasks like crowd localization, we enhance these architectures by incorporating a UNet (Ronneberger et al., 2015) module. The list of algorithms we evaluate includes ERM (baseline), CORAL (Sun & Saenko, 2016), DANN (Ganin et al., 2016), MMD (Li et al., 2018), IRM (Arjovsky et al., 2019), SagNet (Nam et al., 2021), VREx (Krueger et al., 2021), Mixup (Zhang et al., 2018), SAM (Foret et al., 2021), EFDM (Zhang et al., 2022), InfoBot (Li et al., 2022), GAM (Zhang et al., 2023b), SAGM (Wang et al., 2023b), CausalIRL (Chevalley et al., 2022), SD (Pezeshki et al., 2021), and DomainDrop (Guo et al., 2023). We then present the results for our proposed method, SemanticHook.

Table 3: $F_1$ scores from multi-source domain training. Columns represent test performance on each domain's test set, with the highest scores in bold. Underlined scores show the best results within single-source domain groups and the best in two-source domain groups that exclude the target domain from training. The Omni means all of domains are included in training.

| Target Domain | Source Domain(s) | | | | | | | | | | | | | | |
|---|---|---|---|---|---|---|---|---|---|---|---|---|---|---|---|
| | Omni | T | S | N | B | TS | TN | TB | SN | SB | NB | TSN | TNB | TSB | SNB |
| T | 61.26 | 62.05 | 58.26 | 40.10 | 11.25 | **62.80** | 61.86 | 61.70 | 56.71 | 56.62 | 40.55 | 62.02 | 61.80 | 61.92 | 56.15 |
| S | 80.48 | 74.69 | 79.40 | 70.30 | 42.95 | 78.57 | 77.92 | 75.09 | 79.70 | 70.70 | 70.82 | **80.71** | 77.94 | 79.95 | 79.22 |
| N | 84.09 | 71.32 | 80.39 | 82.60 | 66.89 | 80.48 | 83.28 | 78.51 | 83.80 | 82.29 | 82.40 | **84.16** | 83.30 | 82.44 | 83.62 |
| B | 85.48 | 62.20 | 71.00 | 81.57 | 83.46 | 72.36 | 82.40 | 84.20 | 82.27 | 84.70 | 85.90 | 82.34 | 84.60 | 84.63 | **85.57** |
| **Avg.** | **77.83** | 67.57 | 72.26 | 68.64 | 51.14 | 73.55 | 76.37 | 74.88 | 75.62 | 73.58 | 69.92 | 77.31 | 76.91 | 77.24 | 76.14 |

As shown in Table 2, ERM, despite being a baseline algorithm, performs competitively against more advanced OOD methods, suggesting that existing approaches may not adequately address the challenges of scale shift generalization. To further investigate this issue from an empirical perspective, we developed SemanticHook, which, while surpassing ERM, shows only marginal improvement.

However, our goal is not to achieve a state-of-the-art algorithm but to create an effective tool for analyzing the scale shift issue. In the following section, we will extend our analysis by answering the following questions:

- Q1: Can scale shift be alleviated by increasing in-distribution data?
- Q2: Can scale distribution be treated as a major attribute in representing crowd images?
- Q3: How does image interpolation influence the scale shift?
- Q4: What components are effective in addressing scale shift (ablation study for SemanticHook)?

### 4.3 EMPIRICAL ANALYSIS

**Q1: Can scale shift be alleviated by increasing in-distribution data?** To address this, we isolate the train, validation, and test sets within each domain in Table 3. By considering all possible domain combinations, we aim to obtain an impartial assessment of generalization. Our analysis begins with single to single generalization, which serves as the baseline case. We find that larger average domain scales correlate with poorer performance, indicating that greater scale shifts diminish generalization. When we increase the in-distribution data through multi-source scenarios, we observe that domains farther from the target domain exert less influence on performance. For instance, the performance results for *TN to S* (77.92%) and *TNB to S* (77.94%) show that the improvement from adding the additional domain *B* is minimal. A similar trend is evident in other cases as well. Thus, we conclude that *in-distribution data offers limited help in alleviating scale shift generalization.*

**Q2: Can scale distribution be treated as one of the major attributes in representing crowd images?** To address this question, we design a novel experimental pipeline. When an attribute is a primary factor in representing crowd images, we can select a *small* core set by sampling identically and independently (IID) from the corresponding attribute distribution. A model trained on this core set should achieve comparable or even better performance than one trained on the entire dataset.

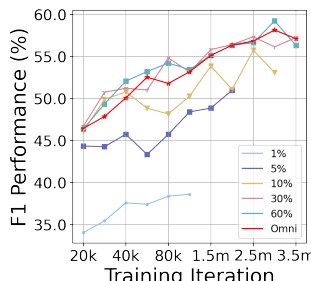

Figure 4: Less is more.

To this end, we IID sample images according to the scale distribution and split them into training, validation, and test sets, ensuring the in-distribution (In-D) property among them. As shown in Figure 4, we observe an intriguing phenomenon: when scale attributes are In-D, we only need 30% of the original dataset to achieve comparable performance. *This further emphasizes that increasing the amount of In-D data provides limited benefits for generalization.*

**Q3: How does image interpolation influence the scale shift?** Theoretically, as discussed in Sec. 3.1, we show that image interpolation can mitigate the shift term of $p(c|y, s, ...)$, but it does not address the shift in $p(s, c, ...)$, thus limiting its effectiveness in dealing with scale shifts. Empirically,

we conduct experiments presented in Table 4 to support this theoretical analysis. Specifically, we use the *Big* domain as our source and generalize it to domains with smaller scales. We implement two strategies involving interpolation: *Random Augmentation* (RA), which randomly interpolates training images; and *Inference Augmentation* (IA), where test images are modified during inference as a form of adversarial attack to assess its impact on model predictions.

As shown in Table 4, the improvement introduced by RA is marginal over Small, Normal, and Big domains. While we assert the improvement on Tiny domain is because its original poor performance. When conducting IA as attack, interpolation still cannot introduce significant influence. We thus derive *while image interpolation provides some relief from scale shift issues, the benefits are modest and more helpful only when the scale shift is significant*.

Table 4: $F_1$ results for different interpolation augmented experiments.

| Interpolation | Tiny | Small | Normal | Big (InD) | Avg |
|---|---|---|---|---|---|
| None | 13.11 | 48.78 | 79.31 | 83.35 | 56.14 |
| RA | 18.72 | 51.38 | 80.73 | 81.00 | 57.96 |
| IA | 12.19 | 44.28 | 73.58 | 81.97 | 53.01 |

**Q4: What kinds of components are effective to scale shift (ablation study for SemanticHook)?**
*Enhance Semantic or Scale Association.* We firstly ablate the type of perturbation conducted in the SemanticHook. Concretelly, we opt for two perturbations conducted on semantic concentrated feature, while another option is on scale concentrated feature. We compare the performance difference in Table 5. As shown, when introducing scale perturbation, it renders model learn stronger scale association, which should be the spurious association that we don't want. As a result, the performance degrades a lot. In contrast, similar results occur between two semantic perturbations.

*Semantic Hook or Global Feature* Then, we ablate the efficacy of semantic hook, by which we compare the results obtained by enhancing the hooked semantic feature with global feature. As shown, we notice semantic feature performs much better than global feature, which further supports that the extracted global feature contain scale associated feature, which hinders the generalization across domain scales.

*Annealing Factor in Extracting Semantic Feature* As aforementioned, there is a coefficient $\gamma$ to adjust the weight of hooked semantic feature. Empirically, the value of $\gamma$ is annealing along the training. Here we conduct ablation on the annealing process, and analyze the behind intuition. Specifically, when $\gamma$ starts from 0, the representation of semantic feature totally depends on $f_E(x + \epsilon)$, this is because at the begining of training, model does not learn too much task information, which means we need a whole representation to the image. As training goes by, the representation to the original image $f_E(x)$ starts to learn more crowd knowledge, a bigger $\gamma$ is helpful for substracting unwanted features.

Table 5: $F_1$ results for the ablation study of SemanticHook

| Ablation | Tiny | Small | Normal | Big | Avg |
|---|---|---|---|---|---|
| Enhance Semantic or Scale Association | | | | | |
| Gaussian Perturb | 59.26 | 85.90 | 88.03 | 81.19 | 78.60 |
| ColorJitter | 60.14 | 85.45 | 87.52 | 80.89 | 78.50 |
| Interpolation | 57.29 | 81.02 | 80.12 | 65.16 | 70.90 |
| Semantic Hook or Global Feature | | | | | |
| Semantic Hook | 59.26 | 85.90 | 88.03 | 81.19 | 78.60 |
| Global Feature | 41.56 | 76.84 | 83.51 | 74.59 | 69.13 |
| Annealing in Extracting Semantic Feature | | | | | |
| w. Anneal | 59.26 | 85.90 | 88.03 | 81.19 | 78.60 |
| wo. Anneal | 59.61 | 85.07 | 83.19 | 64.25 | 73.03 |

## 5 CONCLUSION

We presented *Scale Shift Domain Generalization* with the realm of crowd localization, a new and applicable research direction. In this paper, we built a benchmark on this task called ScaleBench. Extensive experiments on ScaleBench revealed the limitations of existing domain generalization algorithms in addressing scale shift. Through our analysis, we demonstrated scale shift as a joint shift between diversity and correlation shift. Building upon this property, we proposed an algorithm called Semantic Hook to mitigate the issue, and conducted extensive analysis to derive three significant insights for future works. We believe this work serves as a catalyst for greater scholarly attention toward the essential yet challenging task of crowd localization under scale shifts, and we hope it inspires further investigations and advancements in this field.

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

# Appendix

CONTENTS

# A    THEORETICAL PROOF TO THEOREM 1

## A.1    SCALE DISTRIBUTION AND OBJECT DISTRIBUTION

Firstly, we give the definition of scale and object distribution. An object $z$ in an image is defined by its spatial feature and semantic feature. Consequently, understanding the object probability distribution $p(z)$ requires consideration of both the semantic distribution $p(s)$ and the scale distribution $p(c)$, as each influences the model performance.

To study the domain scale shift, we first give a rigorous definition to the scale distribution to crowd localization in Definition 1.

**Definition 1 (Scale Distribution)** *Let variable $z$ represent the object. For one domain with intra-domain scales independently and identically distributed (i.i.d.), the scale distribution can be written as $p(c|z)$, where $c$ denotes the count of pixels occupied by object $z$.*

Based on this definition, we can drive the formula to the object distribution, parameterized by a variable $z$. To begin with the input (pixel value) variable $X$, it obeys a distribution of $X \sim p(x)$, where $p(x)$ is assumed as shared in our setting. Then, the object variable $Z$ is to sample random instances of pixels as $Z = \{X_1, X_2, ..., X_c\}$. Since the $|Z| = c$ is uncertain, we cannot model it via a classic random variable. Thus, we need to introduce the concept of Random Finite Sets (RFS) (Vo et al., 2005) in Definition 2 to model its distribution.

**Definition 2 (Random Finite Sets)** *Let $X$ be the random variable with Probability Density Function (PDF) $p(x)$ defined on a measurable space. The Random Finite Sets (RFS) $Z = \{X_1, X_2, ..., X_c\}$ is defined by the joint distribution of following:*

$$p(z) = \Gamma(c+1) \cdot U^c \cdot p(c) \cdot f_c(x_1, x_2, ..., x_c), \tag{9}$$

*where $U$ is the unit of the hypervolume, $p(c) = Pr(|Z| = c)$ is the cardinality distribution, $f_c$ is the joint distribution over $c$ instances $x$.*

With this definition, we can further simplify it by defining $p(c) \sim \pi(\lambda)$, where $\pi(\lambda)$ denotes a Poisson distribution parameterized by $\lambda$ following (Vo et al., 2005):

$$\begin{aligned} p(z) &= p(c) \cdot \Gamma(c+1) \cdot U^c \cdot f_c(x_1, x_2, ..., x_c) \\ &= \int_{R^\lambda} \frac{e^{-\lambda} \lambda^c}{\Gamma(c+1)} \Gamma(c+1) \cdot U^c \cdot \prod_{i=1}^{c} p(x_i) \cdot p(\lambda) \mathrm{d}\lambda \\ &= U^c \cdot \prod_{i=1}^{c} p(x_i) \int_{R^\lambda} e^{-\lambda} \lambda^c p(\lambda) \mathrm{d}\lambda. \end{aligned} \tag{10}$$

With above derivation, we have the definition to the object distribution $p(z)$.

## A.2    SCALE SHIFT

Following the task setting of out-of-distribution (*OOD*), it is obvious that the scale shift between any two domains can represented as $p_1(c|z) \neq p_2(c|z)$. With this scale shift, we are ready to show that it is a kind of domain shift and how it influences the generalization across domains. To begin with, let us make some formal analysis of the corresponding problem formulation of crowd localization.

**Lemma 1 (Domain Shift (Ye et al., 2022))** *Given scale variable $c$ and output variable $y$, domain shift hinders the generalization of deep model from two aspects:*

$$\begin{cases} \textit{Diversity Shift:} & \exists c \in \mathcal{C} : p_1(c) \cdot p_2(c) = 0, \\ \textit{Correlation Shift:} & \exists y \in \mathcal{Y} : p_1(y|c) \neq p_2(y|c), \end{cases} \tag{11}$$

*where the label shift is not considered here.*

To better facilitate readers understanding to the difference among scale shift, diversity shift, and correlation shift, we illustrate several toy examples in Figure 5. So let us analyze whether $p_1(c|z) \neq$

Figure 5: Toy examples over different kinds of shift.

$p_2(c|z)$ makes influences on any one or more shifts among Eq. 11. Firstly, concentrating on diversity shift, we need to derive a formulation of $p(c)$ from the only known condition of $p(c|z)$. Hence, we make the following derivation:

$$p(c|z) = \frac{p(z|c)p(c)}{p(z)} = \frac{\prod_{i=1}^{c} p(x_i)p(c)}{p(z)} \qquad \text{(Bayes)}$$

$$= \frac{p(c)}{U^c \cdot \int_{R^\lambda} e^{-\lambda} \cdot \lambda^c \cdot p(\lambda)\mathrm{d}\lambda}$$

$$\rightarrow p(c) = p(c|z) \cdot U^c \cdot \int_{R^\lambda} e^{-\lambda} \cdot \lambda^c \cdot p(\lambda)\mathrm{d}\lambda. \qquad (12)$$

According to the last step in Eq. 12, when $p_1(c|z) \neq p_1(c|z)$, let us introduce the diversity shift (Ye et al., 2022) (elaborated in Sec. A.3) formula expression as:

$$\mathrm{Div}_{div}(p_1, p_2) = \frac{1}{2} \int_{R^c} |p_1(c) - p_2(c)|\mathrm{d}c$$

$$= \frac{1}{2} \int_{R^c} \int_{R^\lambda} |p_1(c|z) - p_2(c|z)| \cdot U^c \cdot e^{-\lambda} \cdot \lambda^c \cdot p(\lambda)\mathrm{d}\lambda\mathrm{d}c. \qquad (13)$$

It is obvious that when $p_1(c|z) \neq p_2(c|z)$, $\mathrm{Div}_{div}(p_1(z), p_2(z)) > 0$, which means the existence of diversity shift.

Secondly, let us consider the correlation shift (Ye et al., 2022) (elaborated in Appendix A.3) issue, formulated by:

$$\mathrm{Div}_{cor}(p_1, p_2) = \frac{1}{2} \int_{R^c} \sqrt{p_1(c) \cdot p_2(c)} \sum_{y \in \mathcal{Y}} |p_1(y|c) - p_2(y|c)|\mathrm{d}c$$

$$= \frac{1}{2} \int_{R^c} \sqrt{p_1(c) \cdot p_2(c)} \sum_{y \in \mathcal{Y}} |\frac{p_1(c|y) \cdot p_1(y)}{p_1(c)} - \frac{p_2(c|y) \cdot p_2(y)}{p_2(c)}|\mathrm{d}c. \qquad (14)$$

To verify whether $\mathrm{Div}_{cor}(p_1, p_2)$ exists, we need some further assumptions based on the empirical observation. Commencing from $p(y)$, it can be viewed as the number of objects in each domain. In a real scenario, we observe there is a high correlation between object number with object scale. And this correlation is stable across the dataset. This is because of the fixed image resolution, where one cannot contain many large-scale objects within an image. Therefore, we assume the fraction of $\frac{p(y)}{p(c)}$ is shared across domains. Hence, the $\mathrm{Div}_{cor}(p_1, p_2)$ degenerates into the issue between $p_1(c|y)$ and $p_2(c|y)$.

**Proposition 1 (Linear Expression Among Object Scales)** *For different category objects, the relative scale distribution is fixed across variety of absolute scales. The object scale distributions can be linearly expressed by each other:*

$$\forall m, n \leq |\mathcal{Y}|, \forall i : p_i(c|y_m) = \mathcal{K}_{mn} * p_i(c|y_n), \qquad (15)$$

*where $\mathcal{K}_{mn}$ is the linear kernel defined over $m^{th}$ along with $n^{th}$ class, and is shared among all of domains.*

With Proposition 1, we can draw a generalized conclusion when only considering category *human*. So when $y_{human}$ has $p_1(c|y) \neq p_2(c|y)$, it is clear that Eq. 14 cannot be zero, which proves the existence of correlation shift.

### A.3 SUPPLEMENTARY THEORETICAL DEFINITIONS FOR DOMAIN SHIFT

In summary, we first give a specific formulation to scale shift of $p_1(c|z) \neq p_2(c|z)$. Then, borrowing the definition of the two kinds of generalized shift in Lemma 1, we derive that scale shift can incur diversity shift and correlation shift.

**Definition 3 (Feature Sets)** *To make a decision on $y$ from $x$, there are two kinds of features influencing the process, which are direct cause and confusion, as shown in Fig. 6. For direct cause factors, we have the following equation hold:*

$$p(x) \cdot q(x) \neq 0 \cap \forall y \in \mathcal{Y}: \quad p(y|x) = q(y|x), \tag{16}$$

*where the call these $x$ composed set as $\mathcal{X}_{inv}$. For confusing feature, the opposite property holds:*

$$p(x) \cdot q(x) = 0 \cup \exists y \in \mathcal{Y}: \quad p(y|x) \neq q(y|x), \tag{17}$$

*where the call these $x$ composed set as $\mathcal{X}_{var}$.*

In *OOD* issue, the features $x \in F_{inv}$ should be shared across domains. For any two domains, if $F_{inv} = \emptyset$, we can never successfully make it generalize in these two domains. In a word, $F_{inv} \neq \emptyset$ is the necessary prior to the success of *OOD*.

With these two kinds of feature sets, we can derive the definition to the widely used diversity shift and correlation shift. To begin with, the domain shift can only be shown up in the second feature set $F_{var}$. Coarsely, we can assign the diversity shift to the case when first term in Eq. 17 holds, and the correlation shift to the case when second term holds. Based on this coarse discrimination, we can obtain the definition to diversity shift and correlation shift by partitioning $\mathcal{X}_{var}$.

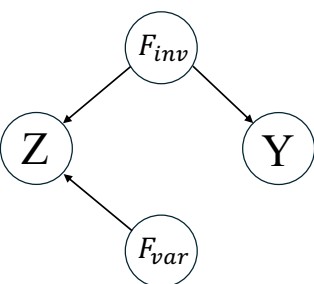

Figure 6: Causal influence among variables.

**Definition 4** *With variant term set $F_{var}$, the diversity shift dominants the scale shift when $x \in \mathcal{S}$, where $\mathcal{S}$ is defined as Eq. 18, and the correlation shift dominants the scale shift when $x \in \mathcal{T}$, which is also defined in Eq. 18.*

$$\mathcal{S} \triangleq \{x \in \mathcal{X}_{var}|p(x) \cdot q(x) = 0\}, \mathcal{D} \triangleq \{x \in \mathcal{X}_{var}|p(x) \cdot q(x) \neq 0\}. \tag{18}$$

**Remark 1** *To understand the two kinds of shifts, let us elaborate them from intuition. Firstly, diversity shift stems from the novel features not shared among domains. So when the $p(x) \neq q(x)$, we can make sure there is novel features in one domain not existing in the other one. Secondly, correlation shift is blamed to the spuriously correlated features with some class. So, given any feature $x$, the object class distribution imbalance incurs the correlation shift, which is namely $p(y|x) \neq q(y|x)$. But to make the formulation more symmetric, we can write it into the form in the second term of Eq. 18.*

Based on the above intuitive remark, we can derive the quantification formula to the diversity shift and correlation shift.

**Lemma 2 (Definition 1 in (Ye et al., 2022))** *Given $\mathcal{S}$ and $\mathcal{T}$ defined in Eq. 18, the diversity shift and correlation can be mathmetically expressed as:*

$$\mathcal{D}_{div}(p,q) \triangleq \frac{1}{2} \int_{\mathcal{S}} |p(x) - q(x)|\mathrm{d}x \tag{19}$$

$$\mathcal{D}_{cor}(p,q) \triangleq \frac{1}{2} \int_{\mathcal{T}} \sqrt{p(x)q(x)} \sum_{y\in\mathcal{Y}} |p(y|x) - q(y|x)|\mathrm{d}x$$

**Lemma 3 (Proposition 1 in (Ye et al., 2022))** *Given two probability distributions $p(x)$ and $q(x)$, which are the corresponding feature distributions in two different domains, the diversity shift $\mathcal{D}_{div}(p,q)$ and correlation shift $\mathcal{D}_{cor}(p,q)$ are always bounded between 0 and 1.*

**Proof 1** *Commencing from the proof to the diversity shift, its upper bound can be easily derived by the triangle inequality:*

$$\mathcal{D}_{div}(p,q) = \frac{1}{2}\int_{\mathcal{S}}|p(x)-q(x)|\mathrm{d}x \leq \frac{1}{2}\int_{\mathcal{S}}[p(x)+q(x)]\mathrm{d}x \leq 1. \tag{20}$$

*Furthermore, we can also prove that in correlation shit as:*

$$
\begin{aligned}
\mathcal{D}_{cor}(p,q) &= \frac{1}{2}\int_{\mathcal{T}}\sqrt{p(x)q(x)}\sum_{y\in\mathcal{Y}}|p(y|x)-q(y|x)|\mathrm{d}x \\
&\leq \frac{1}{2}\int_{\mathcal{T}}\sqrt{p(x)q(x)}\sum_{y\in\mathcal{Y}}|p(y|x)+q(y|x)|\mathrm{d}x \\
&= \frac{1}{2}\int_{\mathcal{T}}2\sqrt{p(x)q(x)}\mathrm{d}x \\
&\leq \frac{1}{2}\int_{\mathcal{T}}[p(x)+q(x)]\mathrm{d}x \\
&\leq 1.
\end{aligned}
\tag{21}
$$

*As for the lower bound, it is obvious based on that the probability cannot be negative.*

$\square$

# B    DETAILS FOR BASELINE CROWD LOCALIZATION MODEL

We illustrate the pipeline of our baseline crowd localization method in Figure 7. Given an image $x \in \mathbb{R}^{H\times W\times 3}$, IIM (Gao et al., 2020), composed of an encoder $f_E$, threshold learner $f_T$ and decoder $f_D$, first embeds $x$ into a feature $f_E(x) \in \mathbb{R}^{\frac{H}{8}\times\frac{W}{8}\times D}$ with a feature dimension of $D$. Then, $f_E(x)$ is fed with threshold learner $f_T$ and decoder $f_D$. Later, the decoder firstly transfers this image embedding into a sigmoid $\sigma$ processed confidence map $\sigma\{f_D[f_E(x)]\} \in [0,1]$, where the $f_D[f_E(x)]$ has a resolution of $(H, W, 1)$, and each value within the map indicates the probability of the corresponding pixel value in the current location belongs to the human head are. Then, this $f_D[f_E(x)]$ is also fed into the threshold along with the aforementioned $f_E(x)$ to obtain a threshold map $T(x) \in [0,1]$, which also has a resolution of $(H, W, 1)$. Then, the final prediction can be obtained by comparing the values by $\mathbb{I}\{f_D[f_E(x)] \geq T(x)\}$, where $\mathbb{I}$ denotes the indicator function.

With this predicted binary map, we can obtain the predicted areas mask. Then, the locations of these foreground areas can be extracted by graphical operation, where the center of each area is treated as the predicted human heads' location.

However, in our task, to make the model concise enough to promise its adaptability, we remove the processes concerning the learnable threshold map, in which we directly obtain a confidence from encoder and decoder, then obtain the final binary map via a global and fixed threshold of 0.5. Experiments in the main text showcases this threshold is able to generalize well under the InD data.

# C    DETAILS FOR REPRODUCED *OOD* ALGORITHMS

In this section, we list the details of our reproduced algorithms.

**Empirical Risk Minimization** (ERM) (Vapnik, 1999): This is the baseline *OOD* algorithm, where the crowd locator is trained in a fully supervised manner on the source domains' training set, then we select the model performs best in the validation set of source domains. With this model, we test its generalization performance on the whole set of target domain.

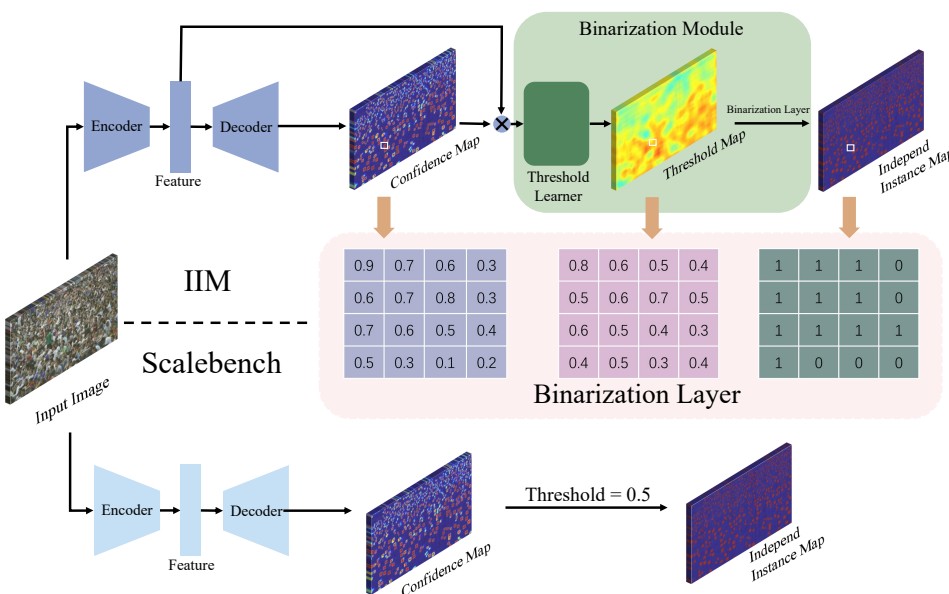

Figure 7: Pipeline for the crowd localization IIM, where we make certain modification to it to make it more concise and generalized enough to be our baseline model.

**Correlation Alignment for Domain Adaptation** (CORAL) (Sun & Saenko, 2016): CORAL minimizes domain shift by aligning the second-order statistics of source and target distributions, without requiring any target labels. In task setting of *OOD*, we conduct CORAL among source domains to learn an invariant features.

**Domain Adversarial Neural Network** (DANN) (Ganin et al., 2016): Based on ERM, DANN has a domain discriminator which aims to enhance the discrimination of predicted feature over domains. This can be viewed as an adversarial paradigm.

**Maximum Mean Discrepancy** (MMD) (Li et al., 2018): In MMD, it tries to minimize the maximum mean discrepancy among source distributions. Empirically, to obtain the feature distribution, we use a Gaussian kernel to transfer it into reproducing kernel Hilbert space (RKHS).

**Invariant Risk Minimization** (IRM) (Arjovsky et al., 2019): IRM seeks to find representations that are invariant across different environments by minimizing a combination of the empirical risk and a penalty term that measures the divergence of optimal predictors across environments. The idea is to make the representation good for all environments simultaneously, which is hypothesized to lead to better *OOD* generalization.

**Manifold Mixup** (Mixup-F) (Verma et al., 2019): Mixup-F is a technique that extends the idea of Mixup-Ito the hidden representations within a neural network. It generates virtual training examples by combining hidden representations of different training examples along with their corresponding labels. This regularizes the neural network to favor simple linear behaviors in-between training examples, which can lead to better generalization.

**Mixup** (Yan et al., 2020): Mixup-Itrains a model on convex combinations of pairs of examples and their labels. By doing this, it encourages the model to behave linearly in-between training examples, which can help to regularize the model and can potentially improve the *OOD* performance by making the model less certain on interpolations between training domains.

**Sharpness Aware Minimization** (SAM) (Foret et al., 2021): SAM seeks to improve model generalization by focusing on the sharpness of the loss landscape. By minimizing the worst-case loss within a neighborhood around the parameters, SAM aims to find parameters that lie in flatter regions of the loss landscape, under the assumption that flatter minima correlate with better generalization, especially in *OOD* scenarios.

**Variance Training Risks** (VREx) (Krueger et al., 2021): VREx is a method that minimizes the variance of the empirical risk across different environments. The intuition is that by finding a model that has stable performance across various source domains, it will likely perform well on unseen target domains.

**Spectral Decoupling** (SD) (Pezeshki et al., 2021): SD addresses the overfitting to spurious correlations by decoupling the spectral components of the feature representations. It does so by regularizing the spectral norm of the weights, which encourages the model to rely less on features that are highly predictive on the training data but may not generalize well to *OOD* data.

**Style-Agnostic Networks** (SagNets) (Nam et al., 2021): SagNets are designed to disentangle content and style information in the neural representations to improve *OOD* generalization. The network learns to separate style-related features from content-related features, and during inference, it relies more on content features, which are presumed to be more stable across different domains.

**Invariant Representation Learning** (IRL) (Chevalley et al., 2022): It bridges the gap between causal reasoning and representation learning. And it establishes a foundation for understanding invariance in the face of style variations.

**Information Bottleneck** (IB) (Li et al., 2022): The IB principle aims at finding a representation that preserves as much information as possible about the target variable while compressing the input data, effectively reducing its complexity. This is achieved by minimizing a trade-off between the mutual information of the representation and the target and the mutual information of the input and the representation. In OOD settings, this can lead to learning more robust features that are less sensitive to variations not relevant to the prediction task.

**Exact Feature Distribution Matching** (EFDM) (Zhang et al., 2022): The EFDM approach is designed to address the limitations of traditional feature distribution matching methods in the context of Arbitrary Style Transfer (AST) and Domain Generalization (DG) tasks. These tasks are predicated on the idea that matching the feature distributions between different domains or styles can improve the performance of visual learning models.

**DomainDrop** (Guo et al., 2023): The DomainDrop approach is an innovative method designed to improve domain generalization, which is the ability of deep neural network models to perform well on unseen test datasets that may have different distributions from the training (source) datasets. The central challenge being addressed is the performance degradation that occurs due to domain shifts, meaning differences between the data distributions of the source and target domains.

**SAGM** (Wang et al., 2023b): SAGM is an optimization method designed to enhance the domain generalization (DG) capabilities of machine learning models. The main goal of DG is to train models on a source domain in such a way that they can perform well when applied to unseen target domains.

**GAM** (Zhang et al., 2023b): GAM is an optimization approach that seeks to enhance the generalization of deep learning models by targeting minima with uniformly small curvature across all directions in the loss landscape. The motivation behind GAM stems from the observed benefits of training models to find *flat* minima—regions of the parameter space where the loss function varies slowly with parameter changes, which are associated with better generalization to unseen data.

## D    EXPERIMENTAL SETTINGS

### D.1    SCALEBENCH GENERATION

In our study, we began by aggregating a comprehensive collection of images. Subsequently, we extracted relevant information on the scale and coordinates of each instance within these images. This data was instrumental in fitting a two-dimensional Gaussian mixture model. Specifically, we normalized the scales by dividing by the maximum scale value and normalized the vertical coordinates by the height of the image. The normalized scale and coordinate data were then combined and inputted into an Expectation-Maximization (EM) algorithm to optimize the parameters of a Gaussian mixture distribution. We preset the number of Gaussian components to five for each image, a number intentionally set to include some redundancy. Following this, we segmented the images into patches based on their respective sub-Gaussian distributions. An initial filtering step was applied

to these patches, eliminating any with a height less than 100 pixels. Moreover, we implemented a variance restriction on the scale of the patches, discarding those with a scale variance greater than twice the mean scale. Once we had obtained a set of clean patches, we moved on to the domain partitioning phase. During this stage, we generated a scale distribution for each patch and employed a greedy search algorithm to identify the optimal scale boundary. This boundary was used to divide the complete scale distribution into five discrete regions. The first four regions were designated for the formation of the ScaleBench, while the fifth region was excluded from further analysis due to its nonconformity with the established criteria.

### D.2 Leave-One-Out Generalization

In our leave-one-out generalization experiments, we conducted a series of four distinct trials. For each trial, one particular domain was designated as the 'target' while the remaining domains collectively formed the 'source' domain. During the training process, we implemented a random rescaling of the input images to vary their size within a range of 0.8 to 1.2 times their original resolution. The images were then randomly cropped to a standard size of $512 \times 512$ pixels. For images with a height smaller than 512 pixels, we employed padding to increase their size to 513 pixels to ensure consistency in input dimensions. Additionally, to augment the dataset and promote model robustness, we included a random horizontal flip for each image.

For algorithms that did not feature a bespoke optimizer, we utilized the Adam optimizer to fine-tune the model parameters. We initiated the optimization with a learning rate of $1 \times 10^{-5}$, which was systematically reduced following each training step at a decay rate of 0.99 to allow for precise adjustments as the model converged. When it came to sampling during training, we tailored our approach to the architecture of each neural network. Specifically, we sampled 8 images for each source domain when training with ResNet-18, 6 images for HRNet, and 4 images for ViT-B. This strategy ensured that each network received an appropriate number of images from the source domains to effectively learn and generalize across the distinct datasets.

### D.3 Multi-Source Generalization

The experimental setup for multi-source generalization mirrors that of the leave-one-out generalization approach in many aspects. However, a key distinction lies in the segregation of the dataset within each domain into three subsets: training (train-set), validation (val-set), and testing (test-set). When a domain is designated as the source domain, both its train-set and val-set are employed for model training and validation, respectively. This allows the model to learn from and tune its parameters based on a diverse range of examples and feedback within the source domain. In contrast, when a domain assumes the role of the target domain, its test-set is exclusively utilized. The performance of the model is then evaluated based on how well it generalizes to this unseen data. This structured approach ensures a clear delineation between the data used for model development and the data used for testing, thereby providing a rigorous assessment of the model's generalization capabilities across different domains.

### D.4 Less is More Experiments

The experimental setting is as same as that in Sec. F.3.

### D.5 Experimental Setting of Table 1

Following official setting of included crowd localization methods, we train their models with corresponding officially released codes on the training set of *Tiny* and *Big* domains. And then, we test each models performance on the test set of *Tiny* and *Big* domains to obtain the results exhibited in Table 1.

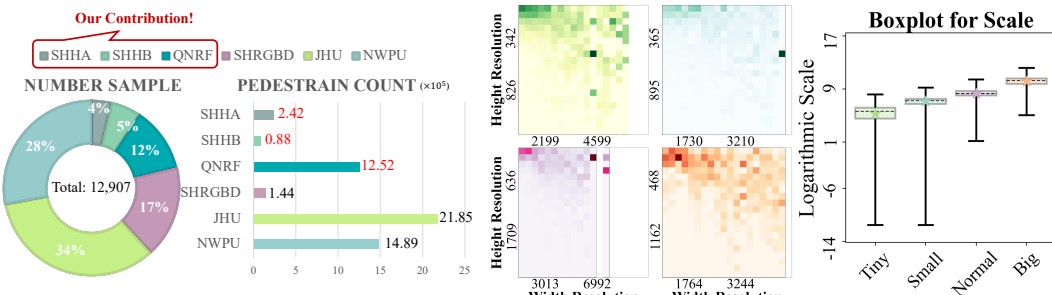

Figure 8: *Left* is the statistic for the number of image and pedestrian within each dataset, where the number of pedestrian we annotated with boxes are in noted in red; *Mid* is the statistic for the resolution distributions; *Right* is the boxplot for the scale distribution, where the star denotes the mean value, the dash line within the box denotes the median value.

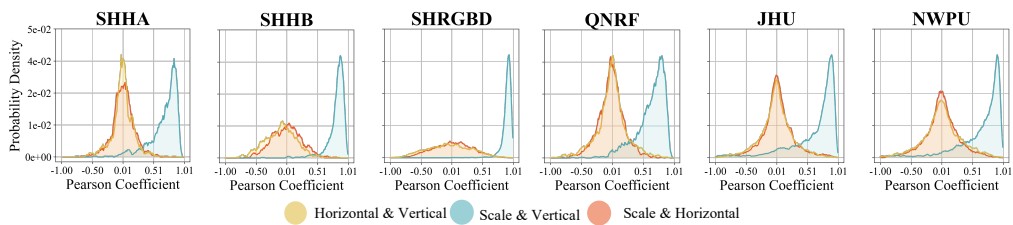

Figure 9: The Pearson correlation value distributions among vertical, horizontal, and scale features.

## E    DATASET STATISTIC INFORMATION

### E.1    DATASETS DISTRIBUTION

The statistical information on included datasets has been depicted in Fig. 8.

### E.2    CORRELATION BETWEEN SCALE WITH VERTICAL AND HORIZONTAL FEATURES.

For domain partitioning, we employ only the vertical features of humans, as per Wang et al. (2023a), to fit Gaussian distributions, guiding the patch-splitting process. We assess the Pearson correlation coefficients among scale, vertical, and horizontal features at the image level and aggregate these correlations across the dataset in Figure 9. Results show a strong positive correlation (close to 1) between scale and vertical features, but no correlation (around 0) with horizontal features. For comparison, the correlation between vertical and horizontal features is also presented, which are irrelevant as a common sense. However, the distribution in scale and horizontal features is closed to that in vertical and horizontal features, which further support our claim.

## F    ADDITIONAL PERFORMANCES

### F.1    LEAVE-ONE-OUT GENERALIZATION

Table 6 presents the leave-one-out test results for the ResNet on the domain *T*, trained on *SNB* domains, and tested on domain *T*. The algorithms are evaluated using six metrics: F1-score, Precision, Recall, Mean Absolute Error (MAE), Mean Square Error (MSE), and Normalized Absolute Error (NAE). Based on the F1-score, the best-performing algorithm is ERM with an F1-score of 23.19, followed by IRM with 23.07. The lowest F1-score belongs to DomainDrop with 6.60. When it comes to Recall, the best performer is again ERM with 13.33, followed by SagNet with 13.26. Overall, ERM and SagNet seem to be the strongest algorithms across most metrics, while DomainDrop has the weakest performance.

Table 7 presents the leave-one-out test results for the ResNet on the domain *S*, trained on *TNB* domains, and tested on domain *S*. The algorithms are evaluated using six metrics: F1-score, Precision, Recall, Mean Absolute Error (MAE), Mean Square Error (MSE), and Normalized Absolute Error (NAE). Based on the F1-score, the best-performing algorithm is SagNet with an F1-score of 69.64, followed by SAGM with 69.48. The lowest F1-score belongs to DomainDrop with 25.92. When it comes to Recall, the best performer is again SagNet with 55.72, followed by SAGM with 54.83. Overall, SAGM and SagNet seem to be the strongest algorithms across most metrics, while DomainDrop has the weakest performance.

Table 8 presents the leave-one-out test results for the ResNet on the domain *N*, trained on *TSB* domains, and tested on domain *N*. The algorithms are evaluated using six metrics: F1-score, Precision, Recall, Mean Absolute Error (MAE), Mean Square Error (MSE), and Normalized Absolute Error (NAE). Based on the F1-score, the best-performing algorithm is SAGM with an F1-score of 80.68, followed by ERM with 79.76. The lowest F1-score belongs to DomainDrop with 56.05. When it comes to Recall, the best performer is again SAGM with 70.11, followed by ERM with 69.42. Overall, ERM and SAGM seem to be the strongest algorithms across most metrics, while DomainDrop has the weakest performance.

Table 9 presents the leave-one-out test results for the ResNet on the domain *B*, trained on *STN* domains, and tested on domain *B*. The algorithms are evaluated using six metrics: F1-score, Precision, Recall, Mean Absolute Error (MAE), Mean Square Error (MSE), and Normalized Absolute Error (NAE). Based on the F1-score, the best-performing algorithm is IB-ERM with an F1-score of 70.11, followed by IB-IRM with 69.64. The lowest F1-score belongs to DomainDrop with 54.14. When it comes to Recall, the best performer is again IB-ERM with 61.19, followed by IB-IRM with 60.18. Overall, IB-ERM and IB-IRM seem to be the strongest algorithms across most metrics, while DomainDrop has the weakest performance.

Table 10 presents the leave-one-out test results for the HRNet on the domain *T*, trained on *SNB* domains, and tested on domain *T*. The algorithms are evaluated using six metrics: F1-score, Precision, Recall, Mean Absolute Error (MAE), Mean Square Error (MSE), and Normalized Absolute Error (NAE). In terms of Precision, GAM leads with a score of 91.25, followed by SAGM with 90.86. Overall, VREx seem to be the strongest algorithms across most metrics, while Mixup-F has the weakest performance.

Table 11 presents the leave-one-out test results for the HRNet on the domain *S*, trained on *TNB* domains, and tested on domain *S*. The algorithms are evaluated using six metrics: F1-score, Precision, Recall, Mean Absolute Error (MAE), Mean Square Error (MSE), and Normalized Absolute Error (NAE). Based on the F1-score, the best-performing algorithm is SAGM with an F1-score of 85.95. The lowest F1-score belongs to Mixup-F with 27.74. In terms of Precision, GAM leads with a score of 95.60, followed by SAM with 95.43. Overall, SAGM seem to be the strongest algorithms across most metrics, while Mixup-F has the weakest performance.

Table 12 presents the leave-one-out test results for the HRNet on the domain *N*, trained on *TSB* domains, and tested on domain *N*. The algorithms are evaluated using six metrics: F1-score, Precision, Recall, Mean Absolute Error (MAE), Mean Square Error (MSE), and Normalized Absolute Error (NAE). The lowest F1-score belongs to Mixup-F with 45.31. In terms of Precision, GAM leads with a score of 96.93, followed by SAM with 96.85. Overall, VREx seem to be the strongest algorithms across most metrics, while Mixup-F has the weakest performance.

Table 13 presents the leave-one-out test results for the HRNet on the domain *B*, trained on *TSN* domains, and tested on domain *B*. The algorithms are evaluated using six metrics: F1-score, Precision, Recall, Mean Absolute Error (MAE), Mean Square Error (MSE), and Normalized Absolute Error (NAE). Based on the F1-score, the best-performing algorithm is VREx with an F1-score of 82.56, followed by CORAL with 82.12. The lowest F1-score belongs to Mixup-F with 19.53. In terms of Precision, SAGM leads with a score of 97.99, followed by SAM with 97.77. When it comes to Recall, the best performer is VREx with 79.43, followed by IRM with 72.95. Overall, VREx and CORAL seem to be the strongest algorithms across most metrics, while Mixup-F has the weakest performance.

Table 14 presents the leave-one-out test results for the ViT-B on the domain *T*, trained on *SNB* domains, and tested on domain *T*. The algorithms are evaluated using six metrics: F1-score, Precision, Recall, Mean Absolute Error (MAE), Mean Square Error (MSE), and Normalized Absolute Error

(NAE). The lowest F1-score belongs to IRL-MMD with 33.08. In terms of Precision, IRL-Gaussian leads with a score of 91.28, followed by GAM with 91.27. Overall, CORAL seem to be the strongest algorithms across most metrics, while IRL-MMD has the weakest performance.

Table 15 presents the leave-one-out test results for the ViT-B on the domain *S*, trained on *TNB* domains, and tested on domain *S*. The algorithms are evaluated using six metrics: F1-score, Precision, Recall, Mean Absolute Error (MAE), Mean Square Error (MSE), and Normalized Absolute Error (NAE). The lowest F1-score belongs to IRL-Gaussian with 73.99. In terms of Precision, SD leads with a score of 94.79, followed by EFDM_Img with 94.00. Overall, CORAL seems to be the strongest algorithms across most metrics, while IRL-Gaussian has the weakest performance.

Table 16 presents the leave-one-out test results for the ViT-B on the domain *N*, trained on *STB* domains, and tested on domain *N*. The algorithms are evaluated using six metrics: F1-score, Precision, Recall, Mean Absolute Error (MAE), Mean Square Error (MSE), and Normalized Absolute Error (NAE). Based on the F1-score, the best-performing algorithm is VREx with an F1-score of 89.91, followed by CORAL with 89.68. The lowest F1-score belongs to IRL-MMD with 76.39. In terms of Precision, SAG leads with a score of 95.73, followed by SD with 95.50. When it comes to Recall, the best performer is again CORAL with 87.77, followed by VREx with 86.10. Overall, VREx and CORAL seem to be the strongest algorithms across most metrics, while IRL-MMD has the weakest performance.

Table 17 presents the leave-one-out test results for the ViT-B on the domain *B*, trained on *STN* domains, and tested on domain *B*. The algorithms are evaluated using six metrics: F1-score, Precision, Recall, Mean Absolute Error (MAE), Mean Square Error (MSE), and Normalized Absolute Error (NAE). Based on the F1-score, the best-performing algorithm is VREx with an F1-score of 83.28, followed by CORAL with 82.79. The lowest F1-score belongs to IRL-Gaussian with 63.21. In terms of Precision, SAGM leads with a score of 97.67, followed by GAM with 97.65. When it comes to Recall, the best performer is again VREx with 74.92, followed by CORAL with 74.56. Overall, VREx and CORAL seem to be the strongest algorithms across most metrics, while IRL-Gaussian has the weakest performance.

Table 6: The leave-one-out results (%) for ResNet on the domain *T*, which is trained on *SNB* domains, and tested on domain *T*.

| Algorithm | Venue | **F1-Score** | Pre. | Rec. | MAE | MSE | NAE |
|---|---|---|---|---|---|---|---|
| ERM | None | **23.19** | 89.09 | **13.33** | **295.69** | 901.80 | **0.69** |
| Coral | ECCV16 | 14.78 | 83.38 | 8.11 | 313.81 | 911.56 | 0.79 |
| DANN | JMLR16 | 20.87 | 87.76 | 11.84 | 300.81 | 905.74 | 0.72 |
| MMD | CVPR18 | 13.14 | 79.61 | 7.16 | 316.39 | 913.81 | 0.81 |
| IRM | arXiv19 | 23.07 | 89.16 | 13.25 | 296.05 | 902.26 | **0.69** |
| Manifold-Mu | ICLR19 | 20.79 | 89.64 | 11.76 | 302.08 | 907.22 | 0.72 |
| Mixup-Img | arXiv20 | 18.60 | 90.01 | 10.37 | 307.57 | 912.45 | 0.75 |
| SAM | ICLR20 | 20.10 | 90.44 | 11.31 | 304.22 | 909.50 | 0.73 |
| VREx | ICML21 | 21.53 | 89.00 | 12.25 | 299.84 | 905.27 | 0.71 |
| SD | NeurIPS21 | 19.94 | 90.59 | 11.21 | 304.64 | 910.53 | 0.73 |
| SagNet | CVPR21 | 23.07 | 88.63 | 13.26 | 295.74 | **901.40** | **0.69** |
| IRL-Gaussian | arXiv22 | 12.34 | 80.12 | 6.69 | 318.58 | 917.60 | 0.82 |
| IRL-MMD | arXiv22 | 12.57 | 80.53 | 6.82 | 318.19 | 916.28 | 0.82 |
| IB-IRM | AAAI22 | 15.30 | 87.18 | 8.39 | 314.21 | 914.55 | 0.79 |
| IB-ERM | AAAI22 | 15.62 | 86.97 | 8.58 | 313.37 | 913.76 | 0.78 |
| EFDM-Feat | CVPR22 | 22.13 | 88.84 | 12.64 | 298.25 | 904.66 | 0.70 |
| EFDM-Img | CVPR22 | 17.91 | 88.59 | 9.96 | 308.54 | 913.16 | 0.75 |
| DomainDrop | ICCV23 | 6.60 | 79.85 | 3.44 | 332.57 | 930.50 | 0.89 |
| SAGM | CVPR23 | 22.78 | 91.03 | 13.02 | 297.94 | 904.64 | 0.70 |
| GAM | CVPR23 | 19.38 | 90.39 | 10.85 | 305.90 | 911.74 | 0.73 |
| Semantic Hook | - | 23.37 | 88.78 | 13.46 | 295.09 | 898.77 | 0.69 |

Table 7: The leave-one-out results (%) for ResNet on the domain *S*, which is trained on *TNB* domains, and tested on domain *S*.

| Algorithm | Venue | F1-Score | Pre. | Rec. | MAE | MSE | NAE |
|---|---|---|---|---|---|---|---|
| ERM | None | 68.27 | 93.42 | 53.79 | 64.32 | 218.30 | 0.35 |
| Coral | ECCV16 | 50.53 | 91.43 | 34.91 | 92.89 | 279.93 | 0.54 |
| DANN | JMLR16 | 63.17 | 93.17 | 47.79 | 73.53 | 236.31 | 0.41 |
| MMD | CVPR18 | 46.09 | 89.59 | 31.02 | 98.40 | 287.56 | 0.59 |
| IRM | arXiv19 | 68.26 | 93.48 | 53.76 | 64.41 | 218.45 | 0.35 |
| Manifold-Mu | ICLR19 | 64.98 | 93.82 | 49.70 | 70.91 | 235.99 | 0.39 |
| Mixup-Img | arXiv20 | 61.48 | 94.43 | 45.58 | 77.76 | 253.15 | 0.42 |
| SAM | ICLR20 | 64.99 | 94.27 | 49.58 | 71.44 | 239.06 | 0.38 |
| VREx | ICML21 | 65.98 | 93.44 | 50.99 | 68.59 | 229.97 | 0.37 |
| SD | NeurIPS21 | 64.58 | 94.51 | 49.05 | 72.34 | 237.71 | 0.39 |
| SagNet | CVPR21 | **69.64** | 92.82 | **55.72** | 60.93 | **208.77** | **0.34** |
| IRL-Gaussian | arXiv22 | 45.70 | 90.05 | 30.62 | 99.21 | 288.08 | 0.59 |
| IRL-MMD | arXiv22 | 45.86 | 89.39 | 30.84 | 98.61 | 292.13 | 0.58 |
| IB-IRM | AAAI22 | 54.82 | 93.06 | 38.85 | 87.63 | 272.20 | 0.48 |
| IB-ERM | AAAI23 | 55.48 | 92.89 | 39.55 | 86.43 | 269.62 | 0.48 |
| EFDM-Feat | CVPR22 | 67.24 | 93.52 | 52.49 | 66.41 | 224.21 | 0.37 |
| EFDM-Img | CVPR22 | 61.00 | 94.31 | 45.08 | 78.48 | 253.14 | 0.42 |
| DomainDrop | ICCV23 | 25.92 | 89.84 | 15.14 | 124.78 | 338.90 | 0.75 |
| SAGM | CVPR23 | 69.48 | 94.80 | 54.83 | 63.64 | 215.38 | **0.34** |
| GAM | CVPR23 | 62.03 | 95.26 | 45.99 | 77.71 | 251.62 | 0.42 |
| Semantic Hook | - | 67.68 | 93.71 | 52.97 | 65.91 | 222.50 | 0.36 |

Table 8: The leave-one-out results (%) for ResNet on the domain *N*, which is trained on *TSB* domains, and tested on domain *N*.

| Algorithm | Venue | F1-Score | Pre. | Rec. | MAE | MSE | NAE |
|---|---|---|---|---|---|---|---|
| ERM | None | 79.76 | 93.71 | 69.42 | 20.84 | 50.16 | **0.27** |
| Coral | ECCV16 | 70.34 | 88.84 | 58.21 | 28.55 | 66.20 | 0.37 |
| DANN | JMLR16 | 74.81 | 93.14 | 62.51 | 26.35 | 61.07 | 0.34 |
| MMD | CVPR18 | 65.30 | 86.26 | 52.54 | 32.73 | 73.72 | 0.43 |
| IRM | arXiv19 | 79.75 | 93.71 | 69.41 | 20.86 | 50.19 | **0.27** |
| Manifold-Mu | ICLR19 | 78.98 | 92.59 | 68.86 | 21.12 | 51.85 | **0.27** |
| Mixup-Img | arXiv20 | 77.40 | 93.16 | 66.21 | 23.32 | 56.98 | 0.29 |
| SAM | ICLR20 | 79.56 | 93.83 | 69.06 | 21.25 | 51.93 | **0.27** |
| VREx | ICML21 | 79.31 | 93.04 | 69.12 | 20.95 | 51.34 | **0.27** |
| SD | NeurIPS21 | 77.94 | 94.59 | 66.28 | 23.62 | 55.74 | 0.30 |
| SagNet | CVPR21 | 78.63 | 93.03 | 68.09 | 21.59 | 51.25 | 0.29 |
| IRL-Gaussian | arXiv22 | 65.62 | 86.77 | 52.76 | 32.73 | 73.80 | 0.42 |
| IRL-MMD | arXiv22 | 65.56 | 87.09 | 52.57 | 32.61 | 73.13 | 0.42 |
| IB-IRM | AAAI22 | 73.49 | 89.93 | 62.13 | 25.96 | 61.90 | 0.32 |
| IB-ERM | AAAI22 | 73.80 | 89.72 | 62.67 | 25.52 | 61.10 | 0.32 |
| EFDM-Feat | CVPR22 | 78.06 | 94.08 | 66.70 | 23.25 | 53.94 | 0.30 |
| EFDM-Img | CVPR22 | 78.40 | 92.95 | 67.79 | 21.89 | 52.04 | 0.28 |
| DomainDrop | ICCV23 | 56.05 | 84.15 | 42.01 | 41.31 | 94.96 | 0.52 |
| SAGM | CVPR23 | **80.68** | 94.98 | **70.11** | **20.74** | **49.46** | **0.27** |
| GAM | CVPR23 | 77.32 | 94.56 | 65.40 | 24.41 | 57.30 | 0.33 |
| Semantic Hook | - | 79.24 | 94.02 | 68.47 | 21.78 | 52.20 | 0.29 |

Table 9: The leave-one-out results (%) for ResNet on the domain *B*, which is trained on *TSN* domains, and tested on domain *B*.

| Algorithm | Venue | F1-Score | Pre. | Rec. | MAE | MSE | NAE |
|---|---|---|---|---|---|---|---|
| ERM | None | 65.03 | 94.35 | 49.62 | 21.50 | 41.18 | 0.49 |
| Coral | ECCV16 | 63.34 | 85.22 | 50.40 | 20.63 | 40.34 | 0.48 |
| DANN | JMLR16 | 64.87 | 90.97 | 50.41 | 20.73 | 39.78 | 0.49 |
| MMD | CVPR18 | 62.14 | 80.94 | 50.42 | 21.00 | 42.66 | 0.49 |
| IRM | arXiv19 | 64.92 | 94.37 | 49.48 | 21.57 | 41.20 | 0.49 |
| Manifold-Mu | ICLR19 | 68.05 | 91.61 | 54.13 | 19.23 | 37.70 | 0.45 |
| Mixup-Img | arXiv20 | 64.63 | 93.17 | 49.47 | 21.49 | 41.39 | 0.49 |
| SAM | ICLR20 | 67.52 | 92.50 | 53.17 | 19.79 | 38.65 | 0.46 |
| VREx | ICML21 | 69.43 | 92.24 | 55.66 | 18.63 | 36.12 | 0.44 |
| SD | NeurIPS21 | 61.37 | 94.85 | 45.36 | 23.47 | 44.20 | 0.53 |
| SagNet | CVPR21 | 62.32 | 92.43 | 47.01 | 22.36 | 42.06 | 0.51 |
| IRL-Gaussian | arXiv22 | 62.54 | 80.79 | 51.02 | 20.63 | 41.73 | 0.48 |
| IRL-MMD | arXiv22 | 62.42 | 80.95 | 50.79 | 20.87 | 41.66 | 0.49 |
| IB-IRM | AAAI22 | 69.64 | 82.62 | 60.18 | 17.10 | 34.91 | **0.41** |
| IB-ERM | AAAI22 | **70.11** | 82.08 | **61.19** | **16.85** | **34.41** | **0.41** |
| EFDM-Feat | CVPR22 | 64.51 | 94.14 | 49.06 | 21.59 | 42.56 | 0.48 |
| EFDM-Img | CVPR22 | 68.75 | 92.52 | 54.69 | 19.06 | 36.99 | 0.44 |
| DomainDrop | ICCV23 | 54.14 | 78.31 | 41.37 | 25.22 | 50.39 | 0.57 |
| SAGM | CVPR23 | 68.25 | 95.07 | 53.23 | 19.89 | 37.61 | 0.47 |
| GAM | CVPR23 | 62.85 | 94.08 | 47.18 | 22.58 | 42.34 | 0.52 |
| Semantic Hook | - | 66.26 | 93.76 | 51.23 | 20.68 | 40.03 | 0.48 |

Table 10: The leave-one-out results (%) for HRNet on the domain *T*, which is trained on *SNB* domains, and tested on domain *T*.

| Algorithm | Venue | F1-Score | Pre. | Rec. | MAE | MSE | NAE |
|---|---|---|---|---|---|---|---|
| ERM | None | 58.05 | 88.74 | 43.13 | 181.45 | 761.96 | 0.34 |
| Coral | ECCV16 | 57.88 | 87.76 | 43.18 | 179.78 | **753.15** | 0.35 |
| DANN | JMLR16 | 39.18 | 89.39 | 25.09 | 250.47 | 852.11 | 0.53 |
| MMD | CVPR18 | 33.47 | 81.76 | 21.04 | 259.25 | 862.77 | 0.57 |
| IRM | arXiv19 | 57.65 | 88.86 | 42.67 | 183.52 | 762.38 | 0.35 |
| Manifold-Mu | ICLR19 | 8.65 | 59.52 | 4.66 | 323.06 | 928.51 | 0.81 |
| Mixup-Img | arXiv20 | 56.05 | 89.40 | 40.83 | 191.04 | 779.26 | 0.37 |
| SAM | ICLR20 | 57.36 | 90.62 | 41.96 | 189.04 | 784.63 | 0.37 |
| VREx | ICML21 | 58.77 | 87.56 | 44.23 | 175.70 | 753.63 | 0.33 |
| SD | NeurIPS21 | 55.40 | 90.01 | 40.02 | 194.95 | 783.06 | 0.38 |
| SagNet | CVPR21 | 57.70 | 88.73 | 42.75 | 182.92 | 767.73 | 0.34 |
| IRL-Gaussian | arXiv22 | 40.67 | 86.39 | 26.60 | 241.84 | 843.23 | 0.51 |
| IRL-MMD | arXiv22 | 41.16 | 86.28 | 27.03 | 239.68 | 844.09 | 0.50 |
| IB-IRM | AAAI22 | 55.50 | 88.89 | 40.35 | 192.65 | 789.34 | 0.37 |
| IB-ERM | AAAI22 | 55.72 | 88.88 | 40.58 | 191.69 | 790.45 | 0.37 |
| EFDM-Feat | CVPR22 | 56.55 | 88.42 | 41.57 | 186.96 | 775.91 | 0.35 |
| EFDM-Img | CVPR22 | 56.83 | 89.18 | 41.70 | 188.08 | 771.54 | 0.36 |
| DomainDrop | ICCV23 | 45.97 | 88.62 | 31.04 | 227.23 | 836.95 | 0.45 |
| SAGM | CVPR23 | 55.15 | 90.86 | 39.59 | 197.61 | 798.25 | 0.38 |
| GAM | CVPR23 | 50.36 | **91.25** | 34.78 | 216.58 | 824.68 | 0.42 |
| Semantic Hook | - | 59.26 | 86.56 | 45.06 | 171.58 | 755.75 | 0.32 |

Table 11: The leave-one-out results (%) for HRNet on the domain *S*, which is trained on *TNB* domains, and tested on domain *S*.

| Algorithm | Venue | F1-Score | Pre. | Rec. | MAE | MSE | NAE |
|---|---|---|---|---|---|---|---|
| ERM | None | 85.30 | 94.04 | 78.05 | 27.56 | 80.89 | 0.17 |
| Coral | ECCV16 | 84.46 | 93.48 | 77.03 | 28.96 | 84.51 | 0.17 |
| DANN | JMLR16 | 74.79 | 93.63 | 62.26 | 51.38 | 164.25 | 0.28 |
| MMD | CVPR18 | 72.70 | 83.98 | 64.09 | 44.79 | 155.64 | 0.28 |
| IRM | arXiv19 | 85.20 | 94.08 | 77.86 | 28.00 | 81.81 | 0.17 |
| Manifold-Mu | ICLR19 | 27.74 | 64.01 | 17.71 | 113.86 | 335.40 | 0.61 |
| Mixup-Img | arXiv20 | 84.64 | 94.21 | 76.84 | 29.87 | 89.10 | 0.17 |
| SAM | ICLR20 | 85.75 | 95.43 | 77.86 | 29.01 | 79.22 | 0.18 |
| VREx | ICML21 | 85.24 | 92.61 | 78.96 | 25.89 | 77.96 | **0.16** |
| SD | NeurIPS21 | 84.21 | 94.77 | 75.77 | 31.34 | 91.40 | 0.18 |
| SagNet | CVPR21 | 85.30 | 93.83 | 78.20 | 27.11 | 80.27 | 0.17 |
| IRL-Gaussian | arXiv22 | 77.76 | 87.62 | 69.89 | 36.97 | 129.80 | 0.23 |
| IRL-MMD | arXiv22 | 77.24 | 87.88 | 68.90 | 38.38 | 132.18 | 0.23 |
| IB-IRM | AAAI22 | 84.52 | 94.89 | 76.19 | 31.10 | 88.13 | 0.19 |
| IB-ERM | AAAI22 | 84.72 | 94.71 | 76.63 | 30.44 | 85.99 | 0.19 |
| EFDM-Feat | CVPR22 | 85.13 | 94.16 | 77.67 | 28.15 | 80.78 | 0.17 |
| EFDM-Img | CVPR22 | 85.04 | 94.10 | 77.57 | 28.37 | 81.98 | 0.18 |
| DomainDrop | ICCV23 | 82.46 | 92.77 | 74.21 | 32.55 | 100.59 | 0.20 |
| SAGM | CVPR23 | **85.95** | 94.86 | 78.56 | 27.79 | 81.42 | 0.17 |
| GAM | CVPR23 | 84.55 | **95.60** | 75.78 | 32.24 | 89.91 | 0.20 |
| Semantic Hook | - | 85.90 | 92.90 | 79.89 | 24.46 | 67.58 | 0.16 |

Table 12: The leave-one-out results (%) for HRNet on the domain *N*, which is trained on *TSB* domains, and tested on domain *N*.

| Algorithm | Venue | F1-Score | Pre. | Rec. | MAE | MSE | NAE |
|---|---|---|---|---|---|---|---|
| ERM | None | 87.90 | 94.06 | 82.50 | 11.13 | 29.25 | **0.16** |
| Coral | ECCV16 | 87.37 | 92.78 | 82.55 | 11.23 | 31.07 | **0.16** |
| DANN | JMLR16 | 81.05 | 93.25 | 71.67 | 19.53 | 47.33 | 0.26 |
| MMD | CVPR18 | 74.37 | 71.73 | 77.22 | 22.94 | 48.97 | 0.39 |
| IRM | arXiv19 | 87.85 | 94.19 | 82.30 | 11.34 | 29.61 | **0.16** |
| Manifold-Mu | ICLR19 | 45.31 | 61.66 | 35.81 | 46.58 | 110.64 | 0.55 |
| Mixup-Img | arXiv20 | 87.71 | 94.82 | 81.59 | 12.10 | 32.68 | 0.17 |
| SAM | ICLR20 | 87.96 | 96.85 | 80.57 | 13.58 | 35.26 | 0.19 |
| VREx | ICML21 | 87.63 | 91.92 | **83.72** | 10.32 | 28.32 | **0.16** |
| SD | NeurIPS21 | 87.22 | 94.73 | 80.81 | 12.57 | 33.45 | 0.17 |
| SagNet | CVPR21 | 87.49 | 93.30 | 82.37 | 11.34 | 30.44 | **0.16** |
| IRL-Gaussian | arXiv22 | 79.97 | 80.39 | 79.57 | 16.00 | 36.22 | 0.26 |
| IRL-MMD | arXiv22 | 79.24 | 79.32 | 79.15 | 16.21 | 37.68 | 0.27 |
| IB-IRM | AAAI22 | 87.02 | 95.87 | 79.67 | 13.86 | 35.58 | 0.19 |
| IB-ERM | AAAI22 | 87.20 | 95.89 | 79.95 | 13.66 | 35.25 | 0.19 |
| EFDM-Feat | CVPR22 | 87.44 | 94.54 | 81.34 | 12.13 | 32.81 | 0.17 |
| EFDM-Img | CVPR22 | 87.60 | 94.68 | 81.51 | 12.17 | 33.05 | 0.17 |
| DomainDrop | ICCV23 | 84.81 | 91.29 | 79.19 | 13.14 | 34.09 | 0.20 |
| SAGM | CVPR23 | 87.91 | 96.60 | 80.64 | 13.50 | 34.22 | 0.19 |
| GAM | CVPR23 | 86.96 | **96.93** | 78.85 | 14.85 | 37.12 | 0.21 |
| Semantic Hook | - | 88.03 | 92.99 | 83.58 | 10.39 | 28.01 | 0.16 |

Table 13: The leave-one-out results (%) for HRNet on the domain *B*, which is trained on *TSN* domains, and tested on domain *B*.

| Algorithm | Venue | F1-Score | Pre. | Rec. | MAE | MSE | NAE |
|---|---|---|---|---|---|---|---|
| ERM | None | 81.57 | 91.62 | 73.50 | 10.13 | 22.07 | 0.25 |
| Coral | ECCV16 | 82.12 | 90.14 | 75.41 | 9.65 | 20.97 | 0.25 |
| DANN | JMLR16 | 73.14 | 90.32 | 61.45 | 15.54 | 32.07 | 0.39 |
| MMD | CVPR18 | 57.01 | 45.80 | 75.48 | 38.87 | 88.29 | 0.88 |
| IRM | arXiv19 | 81.38 | 92.01 | 72.95 | 10.50 | 22.45 | 0.26 |
| Manifold-Mu | ICLR19 | 19.53 | 56.77 | 11.79 | 39.44 | 72.01 | 0.86 |
| Mixup-Img | arXiv20 | 78.69 | 93.90 | 67.72 | 12.92 | 27.07 | 0.31 |
| SAM | ICLR20 | 75.51 | 97.77 | 61.51 | 16.48 | 32.00 | 0.39 |
| VREx | ICML21 | 82.56 | 85.95 | 79.43 | 8.70 | 18.68 | 0.23 |
| SD | NeurIPS21 | 79.98 | 93.33 | 69.98 | 11.83 | 24.87 | 0.29 |
| SagNet | CVPR21 | 79.03 | 88.24 | 71.56 | 11.31 | 24.07 | 0.31 |
| IRL-Gaussian | arXiv22 | 66.81 | 61.74 | 72.78 | 20.62 | 43.93 | 0.50 |
| IRL-MMD | arXiv22 | 67.81 | 64.31 | 71.72 | 18.85 | 39.52 | 0.46 |
| IB-IRM | AAAI22 | 77.54 | 96.77 | 64.69 | 14.87 | 29.61 | 0.34 |
| IB-ERM | AAAI22 | 78.03 | 96.64 | 65.43 | 14.47 | 28.88 | 0.33 |
| EFDM-Feat | CVPR22 | 80.22 | 92.90 | 70.59 | 11.38 | 23.61 | 0.29 |
| EFDM-Img | CVPR22 | 79.69 | 94.18 | 69.07 | 12.32 | 24.85 | 0.30 |
| DomainDrop | ICCV23 | 76.35 | 82.48 | 71.07 | 12.36 | 25.56 | 0.33 |
| SAGM | CVPR23 | 70.57 | 97.99 | 55.15 | 19.38 | 36.52 | 0.46 |
| GAM | CVPR23 | 69.81 | 97.21 | 54.45 | 19.52 | 36.93 | 0.47 |
| Semantic Hook | - | 81.19 | 90.62 | 73.53 | 10.20 | 21.40 | 0.28 |

Table 14: The leave-one-out results (%) for ViT-B on the domain *T*, which is trained on *SNB* domains, and tested on domain *T*.

| Algorithm | Venue | F1-Score | Pre. | Rec. | MAE | MSE | NAE |
|---|---|---|---|---|---|---|---|
| ERM | None | 56.19 | 88.05 | 41.26 | 187.63 | 757.86 | 0.36 |
| Coral | ECCV16 | 56.69 | 85.68 | 42.36 | 182.09 | 757.99 | 0.34 |
| DANN | JMLR16 | 48.73 | 85.57 | 34.07 | 211.94 | 816.21 | 0.40 |
| MMD | CVPR18 | 36.96 | 89.76 | 23.27 | 257.80 | 855.82 | 0.58 |
| IRM | arXiv19 | 56.01 | 87.93 | 41.09 | 188.10 | 750.08 | 0.36 |
| Manifold-Mu | ICLR19 | 49.94 | 89.30 | 34.66 | 214.58 | 814.74 | 0.41 |
| Mixup-Img | arXiv20 | 52.77 | 88.52 | 37.59 | 202.45 | 784.05 | 0.38 |
| SAM | ICLR20 | 52.01 | 89.33 | 36.68 | 207.29 | 806.89 | 0.39 |
| VREx | ICML21 | 56.59 | 87.47 | 41.83 | 185.33 | 747.35 | 0.34 |
| SD | NeurIPS21 | 53.15 | 89.29 | 37.84 | 202.46 | 773.52 | 0.39 |
| SagNet | CVPR21 | 55.13 | 88.19 | 40.10 | 192.81 | 779.02 | 0.35 |
| IRL-Gaussian | arXiv22 | 33.64 | 91.28 | 20.62 | 269.31 | 875.07 | 0.60 |
| IRL-MMD | arXiv22 | 33.08 | 91.19 | 20.20 | 270.79 | 876.63 | 0.60 |
| IB-IRM | AAAI22 | 52.77 | 88.98 | 37.51 | 203.41 | 798.29 | 0.39 |
| IB-ERM | AAAI22 | 52.91 | 88.89 | 37.66 | 202.72 | 797.93 | 0.38 |
| EFDM-Feat | CVPR22 | 46.14 | 88.57 | 31.20 | 227.71 | 851.91 | 0.39 |
| EFDM-Img | CVPR22 | 54.68 | 88.44 | 39.57 | 194.79 | 766.45 | 0.37 |
| DomainDrop | ICCV23 | 42.30 | 88.55 | 27.78 | 240.10 | 844.57 | 0.48 |
| SAGM | CVPR23 | 48.23 | 89.72 | 32.98 | 222.03 | 830.66 | 0.42 |
| GAM | CVPR23 | 38.16 | 91.27 | 24.12 | 256.76 | 881.44 | 0.51 |
| Semantic Hook | - | 56.08 | 87.58 | 41.24 | 187.78 | 763.30 | 0.34 |

Table 15: The leave-one-out results (%) for ViT-B on the domain *S*, which is trained on *TNB* domains, and tested on domain *S*.

| Algorithm | Venue | F1-Score | Pre. | Rec. | MAE | MSE | NAE |
|---|---|---|---|---|---|---|---|
| ERM | None | 85.89 | 93.74 | 79.25 | 26.32 | 81.40 | 0.17 |
| Coral | ECCV16 | 86.50 | 91.12 | 82.33 | 22.00 | 62.23 | 0.17 |
| DANN | JMLR16 | 80.81 | 91.91 | 72.11 | 35.70 | 116.24 | 0.21 |
| MMD | CVPR18 | 76.12 | 90.03 | 65.94 | 44.64 | 147.92 | 0.27 |
| IRM | arXiv19 | 85.92 | 93.90 | 79.18 | 26.40 | 81.76 | 0.17 |
| Manifold-Mu | ICLR19 | 84.12 | 93.33 | 76.56 | 30.40 | 92.49 | 0.19 |
| Mixup-Img | arXiv20 | 85.06 | 93.56 | 77.97 | 28.35 | 93.75 | 0.18 |
| SAM | ICLR20 | 85.39 | 93.66 | 78.47 | 27.99 | 94.41 | 0.18 |
| VREx | ICML21 | 86.50 | 93.10 | 80.78 | 24.23 | 73.08 | 0.16 |
| SD | NeurIPS21 | 84.53 | 94.79 | 76.27 | 31.01 | 98.00 | 0.19 |
| SagNet | CVPR21 | 85.74 | 93.22 | 79.37 | 26.65 | 83.77 | 0.17 |
| IRL-Gaussian | arXiv22 | 73.99 | 92.61 | 61.60 | 51.97 | 159.28 | 0.30 |
| IRL-MMD | arXiv22 | 75.81 | 91.39 | 64.76 | 46.87 | 147.20 | 0.27 |
| IB-IRM | AAAI22 | 85.34 | 92.97 | 78.87 | 26.68 | 83.02 | 0.17 |
| IB-ERM | AAAI22 | 85.44 | 93.02 | 79.00 | 26.46 | 80.81 | 0.17 |
| EFDM-Feat | CVPR22 | 84.95 | 93.45 | 77.86 | 28.76 | 125.89 | 0.17 |
| EFDM-Img | CVPR22 | 85.09 | 94.00 | 77.72 | 28.91 | 94.92 | 0.17 |
| DomainDrop | ICCV23 | 80.31 | 90.76 | 72.01 | 36.42 | 118.88 | 0.23 |
| SAGM | CVPR23 | 84.47 | 93.87 | 76.77 | 30.75 | 103.36 | 0.18 |
| GAM | CVPR23 | 81.53 | 92.98 | 72.58 | 36.68 | 146.40 | 0.22 |
| Semantic Hook | - | 86.28 | 93.87 | 79.83 | 25.36 | 72.58 | 0.16 |

Table 16: The leave-one-out results (%) for ViT-B on the domain *N*, which is trained on *TSB* domains, and tested on domain *N*.

| Algorithm | Venue | F1-Score | Pre. | Rec. | MAE | MSE | NAE |
|---|---|---|---|---|---|---|---|
| ERM | None | 89.39 | 94.64 | 84.69 | 9.99 | 27.27 | 0.15 |
| Coral | ECCV16 | 89.68 | 91.68 | 87.77 | 9.07 | 23.47 | 0.16 |
| DANN | JMLR16 | 84.11 | 93.19 | 76.64 | 15.63 | 37.63 | 0.22 |
| MMD | CVPR18 | 76.55 | 92.13 | 65.48 | 23.66 | 55.35 | 0.29 |
| IRM | arXiv19 | 89.42 | 94.83 | 84.59 | 10.06 | 27.12 | 0.15 |
| Manifold-Mu | ICLR19 | 87.92 | 94.52 | 82.19 | 11.86 | 30.25 | 0.17 |
| Mixup-Img | arXiv20 | 89.35 | 94.06 | 85.09 | 9.78 | 26.17 | 0.15 |
| SAM | ICLR20 | 88.90 | 95.73 | 82.97 | 11.53 | 28.18 | 0.18 |
| VREx | ICML21 | 89.91 | 94.07 | 86.10 | 9.11 | 25.26 | 0.14 |
| SD | NeurIPS21 | 88.51 | 95.50 | 82.48 | 11.70 | 31.15 | 0.16 |
| SagNet | CVPR21 | 89.26 | 94.17 | 84.84 | 10.10 | 27.15 | 0.16 |
| IRL-Gaussian | arXiv22 | 78.34 | 91.67 | 68.39 | 21.56 | 50.98 | 0.28 |
| IRL-MMD | arXiv22 | 76.39 | 93.91 | 64.38 | 25.08 | 58.09 | 0.31 |
| IB-IRM | AAAI22 | 88.78 | 94.80 | 83.48 | 10.87 | 28.11 | 0.16 |
| IB-ERM | AAAI22 | 88.81 | 94.65 | 83.65 | 10.75 | 28.11 | 0.16 |
| EFDM-Feat | CVPR22 | 89.45 | 93.92 | 85.39 | 9.69 | 26.59 | 0.15 |
| EFDM-Img | CVPR22 | 89.35 | 95.15 | 84.21 | 10.36 | 26.31 | 0.16 |
| DomainDrop | ICCV23 | 86.47 | 93.00 | 80.80 | 12.71 | 32.36 | 0.19 |
| SAGM | CVPR23 | 89.31 | 93.64 | 85.36 | 9.93 | 25.53 | 0.16 |
| GAM | CVPR23 | 86.78 | 94.09 | 80.52 | 13.45 | 32.13 | 0.22 |
| Semantic Hook | - | 89.93 | 94.41 | 85.86 | 9.22 | 24.24 | 0.14 |

Table 17: The leave-one-out results (%) for ViT-B on the domain *B*, which is trained on *TSN* domains, and tested on domain *B*.

| Algorithm | Venue | F1-Score | Pre. | Rec. | MAE | MSE | NAE |
|---|---|---|---|---|---|---|---|
| ERM | None | 80.98 | 94.32 | 70.95 | 12.26 | 25.56 | 0.31 |
| Coral | ECCV16 | 82.79 | 93.06 | 74.56 | 10.94 | 22.50 | 0.30 |
| DANN | JMLR16 | 71.52 | 90.83 | 58.98 | 17.10 | 32.45 | 0.46 |
| MMD | CVPR18 | 73.75 | 80.37 | 68.13 | 14.61 | 29.78 | 0.39 |
| IRM | arXiv19 | 80.87 | 94.20 | 70.85 | 12.16 | 25.40 | 0.31 |
| Manifold-Mu | ICLR19 | 81.22 | 95.45 | 70.68 | 11.99 | 23.76 | 0.30 |
| Mixup-Img | arXiv20 | 80.35 | 94.71 | 69.77 | 12.83 | 26.06 | 0.32 |
| SAM | ICLR20 | 74.19 | 97.35 | 59.92 | 17.31 | 33.30 | 0.41 |
| VREx | ICML21 | 83.28 | 93.74 | 74.92 | 10.66 | 23.36 | 0.28 |
| SD | NeurIPS21 | 77.23 | 93.51 | 65.77 | 14.33 | 29.30 | 0.34 |
| SagNet | CVPR21 | 80.32 | 89.41 | 72.90 | 11.62 | 24.17 | 0.33 |
| IRL-Gaussian | arXiv22 | 63.21 | 91.29 | 48.34 | 21.42 | 41.03 | 0.50 |
| IRL-MMD | arXiv22 | 68.13 | 89.01 | 55.19 | 18.23 | 35.96 | 0.44 |
| IB-IRM | AAAI22 | 78.81 | 95.95 | 66.87 | 13.88 | 28.08 | 0.34 |
| IB-ERM | AAAI22 | 80.06 | 96.03 | 68.64 | 13.14 | 26.70 | 0.32 |
| EFDM-Feat | CVPR22 | 82.72 | 94.31 | 73.66 | 11.18 | 23.29 | 0.29 |
| EFDM-Img | CVPR22 | 80.76 | 95.51 | 69.96 | 12.57 | 25.88 | 0.31 |
| DomainDrop | ICCV23 | 76.24 | 93.48 | 64.37 | 14.81 | 29.19 | 0.38 |
| SAGM | CVPR23 | 71.66 | 97.67 | 56.59 | 18.81 | 35.70 | 0.44 |
| GAM | CVPR23 | 63.39 | 97.65 | 46.92 | 23.10 | 43.80 | 0.52 |
| Semantic Hook | - | 80.27 | 95.36 | 69.29 | 12.99 | 26.03 | 0.32 |

Table 18: The results (%) for different domains trained model generalizing to domain *T*.

| HRNetW-48 | F1-Score | Pre. | Rec. | MAE | MSE |
|---|---|---|---|---|---|
| JointTrain | 61.26 | **81.11** | 49.22 | 115.98 | 394.14 |
| From T | 62.05 | 73.01 | 53.96 | **95.34** | 343.17 |
| From S | 58.26 | 73.66 | 48.18 | 111.00 | 370.73 |
| From N | 40.10 | 70.40 | 28.03 | 168.65 | 453.64 |
| From B | 11.25 | 59.94 | 6.20 | 248.56 | 514.47 |
| From SNB | 56.15 | 77.65 | 43.97 | 127.01 | 407.89 |
| From TNB | 61.80 | 78.60 | 50.91 | 110.46 | 380.85 |
| From TSB | 61.92 | 78.77 | 51.01 | 108.04 | 370.37 |
| From TSN | 62.02 | 79.13 | 50.99 | 109.27 | 373.83 |
| From TS | **62.80** | 72.39 | **55.45** | 95.84 | **329.16** |
| From TN | 61.86 | 77.86 | 51.31 | 105.86 | 354.47 |
| From TB | 61.70 | 75.70 | 52.10 | 103.20 | 362.20 |
| From SN | 56.71 | 76.68 | 44.99 | 122.50 | 397.26 |
| From SB | 56.62 | 76.29 | 45.01 | 121.85 | 389.02 |
| From NB | 40.55 | 71.73 | 28.26 | 169.63 | 456.19 |

## F.2 MULTI-SOURCE GENERALIZATION

Table 18 presents the results for different domains, with models trained to generalize to domain T using the HRNet-W48 architecture. The models are also evaluated using the F1-score, Precision (Pre.), Recall, Mean Absolute Error (MAE), and Mean Square Error (MSE). From the F1-score perspective, the best-performing model is the one trained on TS with an F1-score of 62.80, and the lowest is the model trained on B with an F1-score of 11.25. In terms of Precision, the highest score is achieved by the model jointly trained on all domains with a precision of 81.11, while the lowest is again the model trained on B with 59.94. Regarding Recall, the top-performing model is the one trained on TS with a recall of 55.45, and the lowest is the model trained on B with 6.20. For MAE, the lowest (best) score is obtained by the model trained on T with a value of 95.34, which suggests it has the least absolute error. On the other hand, the highest MAE is for the model trained on B with 248.56, indicating it has the highest absolute error in predictions. Looking at MSE, the model trained on TS also performs best, with a score of 329.16, representing the tightest clustering of predictions around the true values. Meanwhile, the highest MSE is found in the model trained on B with a score of 514.47, indicating more significant variance in the predictions. Overall, models trained on TN and T show strong generalizability to domain T across all metrics. The models trained on omni-domain, TSN show moderate performance. From Train present lower generalization performance, suggesting that training solely on T might not be sufficient. The model trained on NB consistently performs the worst across all the metrics, indicating that this domain may be substantially different from domain T, resulting in poor generalization. This indicates that the choice of the training domain has a significant impact on the model's performance on domain T, with closer domains providing better generalization.

Summarizing results exhibited in Table 18, 19, 20, 21, we can observe several generalized phenomenons that: 1) Once the target domain is involved in training, its final performance is greatly enhanced. To explain this, we can see that the present of target domain during training reduces the domain divergence. This also further support our claim on the scale shift influences the final generalization performance. 2) Considering the continual distribution of scale, we can observe that when source domains includes the target domain' s scale scope, even the target domain is absent, its final performance is not very low in generalization. 3) The domain farther to the target domain incurs less influence to the final performance.

Table 19: The results (%) for different domains trained model generalizing to domain *S*.

| HRNetW-48 | F1-Score | Pre. | Rec. | MAE | MSE |
|---|---|---|---|---|---|
| JointTrain | 80.48 | **83.25** | 77.90 | 22.10 | 51.72 |
| From T | 74.69 | 72.98 | 76.49 | 31.31 | 57.35 |
| From S | 79.40 | 80.69 | 78.16 | 25.57 | 56.49 |
| From N | 70.30 | 79.93 | 62.74 | 42.71 | 133.11 |
| From B | 42.95 | 74.76 | 30.13 | 85.10 | 216.22 |
| From SNB | 79.22 | 83.07 | 75.70 | 25.70 | 63.91 |
| From TNB | 77.94 | 80.24 | 75.76 | 26.30 | 66.26 |
| From TSB | 79.95 | 82.10 | 77.91 | 22.43 | 49.05 |
| From TSN | **80.71** | 82.92 | 78.61 | **21.82** | 48.37 |
| From TS | 78.57 | 75.95 | **81.38** | 26.03 | **47.31** |
| From TN | 77.92 | 80.55 | 75.45 | 27.79 | 68.81 |
| From TB | 75.09 | 77.20 | 73.09 | 30.27 | 72.78 |
| From SN | 79.70 | 82.60 | 76.90 | 24.20 | 56.90 |
| From SB | 70.70 | 80.50 | 63.00 | 43.10 | 131.00 |
| From NB | 70.82 | 79.57 | 63.80 | 41.91 | 128.05 |

Table 20: The results (%) for different domains trained model generalizing to domain *N*.

| HRNetW-48 | F1-Score | Pre. | Rec. | MAE | MSE |
|---|---|---|---|---|---|
| JointTrain | 84.09 | **86.65** | 81.67 | 13.38 | 33.26 |
| From T | 71.32 | 69.80 | 72.90 | 23.47 | 43.50 |
| From S | 80.39 | 81.23 | 79.57 | 16.90 | 38.14 |
| From N | 82.60 | 83.89 | 81.34 | 13.64 | 34.05 |
| From B | 66.89 | 80.68 | 57.13 | 33.20 | 84.94 |
| From SNB | 83.62 | 86.20 | 81.18 | 13.72 | 34.81 |
| From TNB | 83.30 | 84.00 | **82.62** | 13.16 | **30.47** |
| From TSB | 82.44 | 84.64 | 80.35 | 14.83 | 36.81 |
| From TSN | **84.16** | 85.87 | 82.52 | **12.60** | 30.61 |
| From TS | 80.48 | 81.28 | 79.69 | 17.27 | 38.15 |
| From TN | 83.28 | 85.04 | 81.58 | 13.16 | 31.73 |
| From TB | 78.51 | 81.42 | 75.79 | 17.33 | 45.25 |
| From SN | 83.80 | 85.60 | 82.00 | 13.30 | 32.00 |
| From SB | 82.29 | 84.29 | 80.38 | 15.44 | 38.75 |
| From NB | 82.40 | 85.90 | 79.20 | 14.50 | 38.00 |

Table 21: The results (%) for different domains trained model generalizing to domain *B*.

| HRNetW-48 | F1-Score | Pre. | Rec. | MAE | MSE |
|---|---|---|---|---|---|
| JointTrain | 85.48 | **84.79** | 86.18 | 7.33 | 13.99 |
| From T | 62.20 | 57.16 | 68.22 | 21.01 | 37.13 |
| From S | 71.00 | 63.93 | 79.84 | 19.39 | 31.38 |
| From N | 81.57 | 78.50 | 84.89 | 9.38 | 14.47 |
| From B | 83.46 | 81.34 | 85.68 | 9.33 | 17.84 |
| From SNB | 85.57 | 84.27 | 86.92 | **6.85** | **11.81** |
| From TNB | 84.60 | 80.77 | **88.81** | 8.65 | 14.06 |
| From TSB | 84.63 | 83.65 | 85.63 | 7.96 | 15.38 |
| From TSN | 82.34 | 81.25 | 83.47 | 8.27 | 14.01 |
| From TS | 72.36 | 66.91 | 78.77 | 16.38 | 25.07 |
| From TN | 82.40 | 79.41 | 85.62 | 9.23 | 14.02 |
| From TB | 84.20 | 82.40 | 85.90 | 7.70 | 14.20 |
| From SN | 82.27 | 80.66 | 83.95 | 8.74 | 14.97 |
| From SB | 84.70 | 82.50 | 87.00 | 7.90 | 12.60 |
| From NB | **85.90** | 84.40 | 87.40 | 7.10 | 14.00 |

### F.3 PRE-TRAINED MODELS

To harness the capabilities of vision transformers for our task, we began by extracting the image encoder from the selected vision transformer models. The encoder serves as a feature extractor, capturing the intricate patterns and high-level representations within the images. Recognizing the need to reconstruct the spatial detail lost during the encoding process, we complemented the encoder with a series of transposed convolutional modules. These modules, often referred to as deconvolutional layers, function in a manner inverse to that of standard convolutional layers. By employing learnable filters and strides, they progressively upsample the lower-resolution feature maps output by the encoder, thereby regaining the original spatial resolution of the input images. This architecture, combining the discriminative power of vision transformer encoders with the spatial recovery ability of transposed convolutional modules, forms a robust foundation for our model. However, to fully exploit this structure, we tailored the fine-tuning stage to optimally adjust the pre-trained weights to our specific application. Fine-tuning was performed with a bifurcated learning rate strategy aimed at balancing stability and adaptability. For the decoder, we adopted a learning rate of $1 \times 10^{-5}$, which is relatively higher to encourage the model to learn the nuances of upsampling and reconstruction more rapidly. This rate allows the decoder to adapt to the task of restoring image details without major restrictions. In contrast, for the encoder—which already possesses a wealth of pre-trained knowledge—a much lower learning rate of $1 \times 10^{-8}$ was chosen. This conservative learning rate ensures that the valuable encoded representations are retained and only subtly modified, preventing the overwriting of useful features developed through pre-training on large and diverse datasets. By employing this dual learning rate approach, we strike a delicate balance: we maintain the integrity of the encoder's pre-trained features while allowing the decoder to evolve and specialize for the task at hand. This fine-tuning methodology is designed to bring the entire model in line with the specific requirements of our application, thus enabling the production of high-fidelity results in recovering the original image resolution.

To undertake a comprehensive evaluation of the generalization capabilities of various pre-trained models under conditions of scale shift, we must establish an experimental framework that presents models with a range of object scales that change dynamically.

This framework is essential for simulating scenarios akin to real-world applications where object sizes can differ significantly from those seen during the training phase. The experimental setup is as follows:

1. Creation of a Varied Scale Dataset: Our first step is to assemble a dataset that is reflective of diverse object scales. This dataset is composed of image patches, each containing objects of different sizes. By covering a broad spectrum of scales, the dataset ensures that the scale distribution is continuous and representative of potential real-world variations.

2. Discretization into Scale Bins: With the dataset in hand, we segment the continuous scale distribution into discrete intervals called scale bins. Each bin corresponds to a specific range of object scales and effectively represents a mini-dataset or 'domain.' This segmentation allows us to handle the scale variation in a structured manner, facilitating separate analysis and modeling for each bin.

3. Expansion of Domain Variety: This binning approach stands in contrast to our primary method, referred to as ScaleBench, which limited the domain count to four. By increasing the number of bins, we correspondingly increase the number of distinct domains, thereby enriching the diversity of scale shifts that we can analyze. This methodological shift enables a more detailed investigation into how models respond to subtle changes in scale, beyond the coarse groupings used previously.

4. Domain Partitioning for Training and Testing: We then strategically divide the scale bins into two groups based on a predetermined scale threshold. The bins below this threshold are designated for training the models and selecting the optimal model configurations (modes). Conversely, the bins that exceed this threshold are reserved for testing. This division is critical for evaluating OOD generalization: during training, models are exposed only to a restricted range of scales, while during testing, they encounter scales that they have not been trained on—mirroring the challenges models face when deployed in the real world.

Table 22: Generalization performance (%) of CLIP pre-trained ViT.

| Index | 0 | 20 | 40 | 60 | 80 | 100 | 120 | 140 | 160 | 180 | 200 | 220 |
|---|---|---|---|---|---|---|---|---|---|---|---|---|
| F1-Score | 65 | 60 | 61 | 57 | 54 | 48 | 57 | 49 | 52 | 53 | 47 | 51 |
| Pre. | 63 | 56 | 59 | 48 | 50 | 42 | 44 | 42 | 40 | 41 | 40 | 38 |
| Rec. | 75 | 75 | 76 | 81 | 77 | 70 | 87 | 76 | 83 | 81 | 69 | 80 |

Table 23: Generalization performance (%) of DINO-v2 pre-trained ViT.

| Index | 0 | 20 | 40 | 60 | 80 | 100 | 120 | 140 | 160 | 180 | 200 | 220 |
|---|---|---|---|---|---|---|---|---|---|---|---|---|
| F1-Score | 56 | 54 | 57 | 48 | 53 | 43 | 56 | 45 | 49 | 49 | 44 | 50 |
| Pre. | 67 | 62 | 69 | 51 | 60 | 46 | 48 | 39 | 51 | 46 | 46 | 44 |
| Rec. | 56 | 56 | 61 | 54 | 60 | 57 | 76 | 61 | 65 | 64 | 53 | 61 |

Figure 10 illustrates the variations in model performance in relation to the index of the test domain. This index serves as a proxy for the degree of scale shift—the higher the index, the greater the scale deviation from the training set. The graph provides a visual representation of how each model's accuracy fluctuates in response to progressively larger scale shifts. To complement the visual analysis, we have compiled extensive numerical data, which is detailed in the subsequent tables. These tables present a comprehensive view of the performance metrics—such as accuracy, precision, recall, and F1 score—for each of the scale bins. By dissecting the models' performance across these metrics, we can draw nuanced insights into their resilience and adaptability to varying scale conditions. This elaborate setup and the ensuing detailed analysis enable us to identify

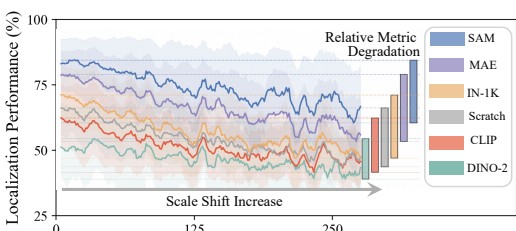

Figure 10: After training crowd locators on a source domain, we evaluated their performance on continuous domains with increasing scale shifts. To better support the results, we utilized various backbone models Deng et al. (2009); Dosovitskiy et al. (2021); Radford et al. (2021); Kirillov et al. (2023); Oquab et al. (2023). See Appendix for detailed experimental setting.

which models are best equipped to maintain high levels of performance across scale shifts. Such models would be particularly advantageous in applications where robustness to scale variation is of paramount importance, such as in surveillance, autonomous driving, or medical imaging, where the ability to accurately recognize objects of varying sizes can be critical to the system's success and reliability.

Table 24: Generalization performance (%) of MAE pre-trained ViT.

| Index | 0 | 20 | 40 | 60 | 80 | 100 | 120 | 140 | 160 | 180 | 200 | 220 |
|---|---|---|---|---|---|---|---|---|---|---|---|---|
| F1-Score | 80 | 75 | 74 | 71 | 70 | 65 | 74 | 61 | 66 | 69 | 69 | 55 |
| Pre. | 81 | 71 | 74 | 65 | 68 | 65 | 68 | 68 | 61 | 66 | 62 | 61 |
| Rec. | 81 | 82 | 81 | 82 | 76 | 72 | 85 | 63 | 78 | 80 | 82 | 65 |

Table 25: Generalization performance (%) of ViT training from scratch.

| Index | 0 | 20 | 40 | 60 | 80 | 100 | 120 | 140 | 160 | 180 | 200 | 220 |
|---|---|---|---|---|---|---|---|---|---|---|---|---|
| F1-Score | 70 | 63 | 66 | 59 | 56 | 54 | 60 | 44 | 55 | 59 | 44 | 51 |
| Pre. | 69 | 59 | 65 | 56 | 53 | 52 | 49 | 34 | 46 | 53 | 40 | 38 |
| Rec. | 75 | 75 | 78 | 75 | 73 | 69 | 87 | 68 | 76 | 80 | 64 | 82 |

Table 26: Generalization performance (%) of SeAM pre-trained ViT.

| Index | 0 | 20 | 40 | 60 | 80 | 100 | 120 | 140 | 160 | 180 | 200 | 220 |
|---|---|---|---|---|---|---|---|---|---|---|---|---|
| F1-Score | 84 | 82 | 80 | 80 | 75 | 71 | 80 | 63 | 79 | 72 | 74 | 63 |
| Pre. | 89 | 82 | 86 | 79 | 79 | 72 | 80 | 75 | 81 | 68 | 77 | 70 |
| Rec. | 82 | 84 | 79 | 82 | 75 | 72 | 84 | 60 | 81 | 79 | 74 | 65 |

Table 27: Generalization performance (%) of ImageNet pre-trained ViT.

| Index | 0 | 20 | 40 | 60 | 80 | 100 | 120 | 140 | 160 | 180 | 200 | 220 |
|---|---|---|---|---|---|---|---|---|---|---|---|---|
| F1-Score | 73 | 68 | 68 | 65 | 62 | 55 | 63 | 51 | 58 | 58 | 54 | 48 |
| Pre. | 73 | 63 | 64 | 60 | 59 | 49 | 56 | 58 | 48 | 49 | 45 | 47 |
| Rec. | 79 | 79 | 79 | 79 | 73 | 74 | 81 | 65 | 79 | 79 | 73 | 76 |

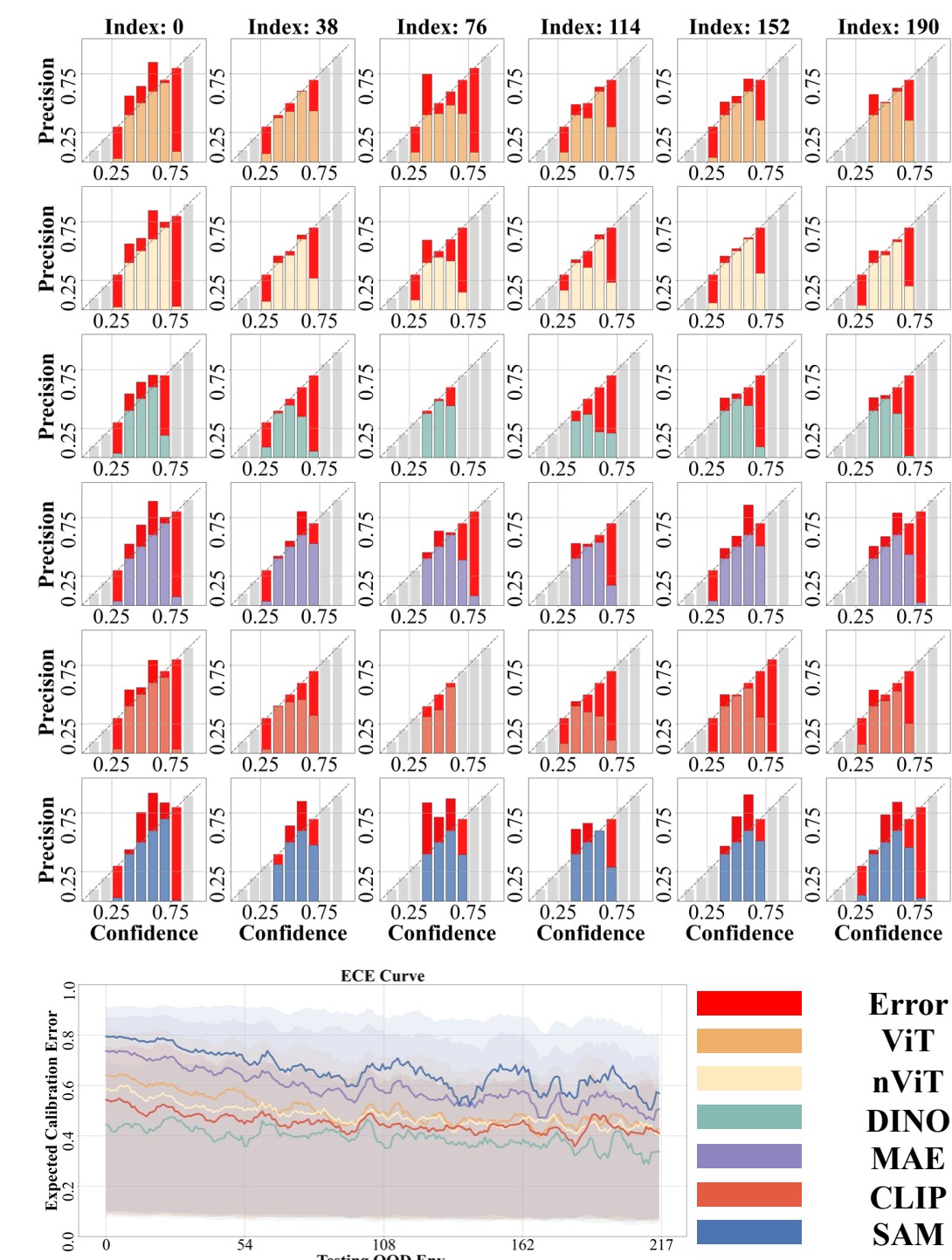

Figure 11: Expected calibration error with different pre-trained vision transformers.

## F.4 CALIBRATION EXPERIMENTS

The relationship between model calibration and out-of-distribution (OOD) generalization performance has become a focal point of investigation within the OOD research community. This interest is driven by the hypothesis that well-calibrated models, which provide accurate probability estimates of their predictions, are also likely to demonstrate better generalization to data that differs from the

distribution seen during training. This concept has been extensively explored and discussed in seminal works within the field. To contribute to this body of research, we propose a novel approach by adapting the notion of calibration to the specific task of crowd localization. Our methodological framework is defined as follows:

**Definition 5** *Consider a set of $N_{pre}$ predicted independent entities, such as individuals in a crowd, as identified by a trained model on an image. For each predicted entity, we ascertain its associated prediction confidence by computing the mean value of the pixels that lie within the ground-truth bounding box on a confidence map—a spatial representation of prediction confidence levels across the image. These predictions are subsequently grouped into 10 equidistant confidence bins, represented as $\{\mathrm{conf}_i\}_{i=1}^{10}$, where each bin spans a confidence interval of 0.1. Within each bin, we derive the bin-specific posterior precision $\{\mathrm{pre}_i\}_{i=1}^{10}$. We then define the expected calibration error (ECE) in a quantitative manner:*

$$\mathrm{ECE} = \sum_{i=1}^{10} \frac{N_i}{N_{pre}} |\mathrm{conf}_i - \mathrm{pre}_i|, \tag{22}$$

*with $N_i$ denoting the number of predictions falling within the $i^{th}$ bin. The ECE serves as a statistical measure of calibration quality, indicating the discrepancy between predicted confidences and actual accuracies.*

In our investigation, we extend the analysis to evaluate how the calibration performance of machine learning models holds up under conditions where the scale of objects in images is subject to variations—a scenario referred to as scale shift generalization. Figure 11 in our paper depicts the calibration errors of six different pre-trained models, each subjected to varying degrees of scale shift. These shifts are indexed, with higher index values signifying more pronounced deviations from the scale of objects seen during training. Our findings reveal a counterintuitive phenomenon: calibration seems to improve with greater scale shifts. This could be attributed to models exhibiting lower confidence in their predictions as the deviation from trained object scales increases—a behavior that may inadvertently lead to better-calibrated predictions. Upon a comparative assessment of various pre-trained models, it becomes evident that the Vision Transformer (ViT) stands out for its calibration accuracy. The ViT's strong performance suggests that its architecture may be inherently more adept at maintaining reliable probabilistic outputs, even in the face of significant scale variations that are characteristic of OOD data. This insight underscores the potential of ViT models for deployment in applications where encountering OOD scenarios is likely, thereby demanding models that can not only generalize well but also provide trustworthy predictions.

### F.5 LESS IS MORE EXPERIMENTS

We specifically explore whether adhering to the feature distribution of the dataset, in this case, the scale distribution, offers a pathway to identifying the smallest yet optimal subset of data—a 'coreset'—that maximizes model performance. This approach is insightful for understanding data economy in the training process. Our experimental design is centered on the distribution of object scales within our dataset. We embarked on an exploration to determine whether a subset of data that mirrors the original scale distribution could lead to efficient model training. Here is the revised and elaborate methodology:

1. Dataset Analysis: We begin with a thorough analysis of the scale distribution within the complete dataset. This involves identifying the range and frequency of object scales present in the dataset, providing a comprehensive overview of the scale feature distribution.

2. Subset Construction: Leveraging this distribution, we construct a subset of the dataset. The selection of data points for this subset is guided by the aim of maintaining the same proportional representation of scale ranges as in the full dataset. This method ensures that the subset is a scaled-down yet faithful microcosm of the original data in terms of scale distribution.

3. Proportional Split into Data Splits: This carefully constructed subset is further split into training, validation (val-), and testing (test-) sets, maintaining the proportional representation of the scale distribution in each split. The proportionality is critical to ensure that the scale variance is consistently represented across all phases of model training and evaluation.

Table 28: Training a ViT on 5% InD scale data. (%)

| Iteration | 20k | 30k | 40k | 50k | 80k | 100k | 150k | 200k |
|-----------|-----|-----|-----|-----|-----|------|------|------|
| F1-Score | 44 | 44 | 46 | 49 | 51 | 43 | 46 | 48 |
| Pre. | 44 | 59 | 54 | 45 | 52 | 77 | 66 | 43 |
| Rec. | 45 | 35 | 40 | 53 | 50 | 30 | 35 | 56 |
| MAE | 146.53 | 121.13 | 132.11 | 152.66 | 131.31 | 116.59 | 119.53 | 169.35 |
| MSE | 448.38 | 439.30 | 442.39 | 444.14 | 425.80 | 437.51 | 4453 | 455.32 |
| NAE | 1.18 | 67 | 1.07 | 1.52 | 1.30 | 55 | 69 | 1.84 |

Table 29: Training a ViT on 10% InD scale data. (%)

| Iter. | 20k | 30k | 40k | 50k | 80k | 100k | 150k | 200k | 250k | 300k |
|-------|-----|-----|-----|-----|-----|------|------|------|------|------|
| F1-Score | 46 | 50 | 51 | 49 | 48 | 50 | 54 | 51 | 56 | 53 |
| Pre. | 54 | 63 | 56 | 68 | 79 | 76 | 57 | 79 | 59 | 77 |
| Rec. | 40 | 41 | 46 | 38 | 35 | 38 | 51 | 38 | 53 | 41 |
| MAE | 120.05 | 107.35 | 120.82 | 111.36 | 109.22 | 105.81 | 119.42 | 101.54 | 110.33 | 97.35 |
| MSE | 427.24 | 414.07 | 424.94 | 430.06 | 435.14 | 429.28 | 415.78 | 424.70 | 403.82 | 413.57 |
| NAE | 0.78 | 0.61 | 0.84 | 0.56 | 0.48 | 0.48 | 1.00 | 0.44 | 0.94 | 0.45 |

4 Efficacy Evaluation: We then engage in training models using this scale-distributed subset and evaluate their performance. The intriguing findings of this experiment are illustrated in Fig. 4. We discovered that by using only 30% of the data, which is proportionally representative of the original scale distribution, the models can achieve performance that is comparable to, and in some cases even slightly surpasses, the performance attained when using the entire dataset (100% of samples). This is a remarkable demonstration of the 'less is more' principle, where the judicious selection of data based on feature distribution can lead to equally or more effective model training.

The implications of these findings are significant. They suggest that an optimal coreset can be accessed by sampling data according to its feature distribution—here, the scale distribution. This methodological insight could lead to substantial computational savings and efficiency improvements in model training, particularly in applications where data is abundant but resources are limited. It also highlights the potential for strategic data selection to enhance the focus of model training on critical features, potentially improving model robustness and reducing overfitting to non-essential data variations. Our results underscore the importance of scale as a determinant of data efficacy in model training. This has profound implications for fields such as computer vision, where scale invariance is a known challenge. By optimizing data selection for scale representation, we can make progress toward more efficient and effective machine learning practices that better leverage the available data.

### F.6 IMAGE INTERPOLATION EXPERIMENTS

In our study, we examined the potential of image interpolation as a countermeasure to address the scale shift effects that often pose challenges in image recognition tasks. Our experimental design centered on the 'Big' domain—which served as our source dataset—and we sought to evaluate the model's ability to generalize this knowledge to various 'Left' domains. These Left domains encompass a range of datasets, including those with images of different resolutions and scales, challenging the robustness and adaptability of our model. To tackle the issue of scale variability,

Table 30: Training a ViT on 30% InD scale data. (%)

| Iter. | 20k | 30k | 40k | 50k | 80k | 100k | 150k | 200k | 250k | 300k | 350k | 400k |
|-------|-----|-----|-----|-----|-----|------|------|------|------|------|------|------|
| F1-Score | 47 | 51 | 51 | 51 | 55 | 53 | 56 | 56 | 57 | 56 | 57 | 58 |
| Pre. | 52 | 58 | 67 | 74 | 74 | 80 | 81 | 79 | 78 | 79 | 79 | 77 |
| Rec. | 42 | 45 | 42 | 39 | 44 | 40 | 43 | 44 | 45 | 44 | 45 | 46 |
| MAE | 126.15 | 113.75 | 107.60 | 104.18 | 99.30 | 101.24 | 94.89 | 95.23 | 94.34 | 97.36 | 93.77 | 94.00 |
| MSE | 427.91 | 413.44 | 419.78 | 422.22 | 410.45 | 417.45 | 411.59 | 411.81 | 408.96 | 412.44 | 411.61 | 405.11 |
| NAE | 0.83 | 0.69 | 0.58 | 0.50 | 0.48 | 0.46 | 0.41 | 0.41 | 0.43 | 0.43 | 0.42 | 0.44 |

Table 31: Training a ViT on 60% InD scale data. (%)

| Iter. | 20k | 30k | 40k | 50k | 80k | 100k | 150k | 200k | 250k | 300k | 350k | 400k | 450k | 500k |
|---|---|---|---|---|---|---|---|---|---|---|---|---|---|---|
| F1-Score | 46 | 49 | 52 | 53 | 54 | 53 | 55 | 56 | 57 | 59 | 56 | 59 | 59 | 58 |
| Pre. | 58 | 62 | 64 | 69 | 77 | 80 | 80 | 83 | 81 | 80 | 83 | 81 | 81 | 83 |
| Rec. | 39 | 41 | 44 | 43 | 42 | 40 | 42 | 43 | 43 | 47 | 43 | 47 | 47 | 45 |
| MAE | 116.82 | 113.54 | 107.67 | 104.38 | 99.39 | 100.38 | 98.01 | 94.72 | 96.20 | 89.78 | 96.56 | 89.99 | 90.40 | 91.75 |
| MSE | 423.80 | 420.24 | 411.62 | 411.13 | 414.90 | 418.92 | 415.59 | 408.22 | 408.90 | 399.07 | 411.61 | 398.37 | 401.14 | 402.15 |
| NAE | 0.66 | 0.63 | 0.63 | 0.55 | 0.46 | 0.45 | 0.43 | 0.41 | 0.42 | 0.41 | 0.42 | 0.40 | 0.39 | 0.39 |

Table 32: Training a ViT on omni-InD scale data. (%)

| Iter. | 20k | 30k | 40k | 50k | 80k | 100k | 150k | 200k | 250k | 300k | 350k | 400k | 450k | 500k | 550k | 600k |
|---|---|---|---|---|---|---|---|---|---|---|---|---|---|---|---|---|
| F1-Score | 46 | 48 | 50 | 52 | 52 | 53 | 55 | 56 | 57 | 58 | 57 | 59 | 57 | 58 | 58 | 60 |
| Pre. | 55 | 64 | 67 | 68 | 76 | 79 | 80 | 83 | 82 | 79 | 85 | 81 | 84 | 82 | 81 | 81 |
| Rec. | 40 | 38 | 40 | 43 | 39 | 40 | 42 | 43 | 43 | 46 | 43 | 46 | 44 | 45 | 45 | 48 |
| MAE | 119.16 | 113.82 | 110.82 | 105.13 | 105.98 | 101.55 | 99.59 | 95.50 | 95.49 | 92.50 | 94.87 | 92.39 | 93.98 | 93.33 | 94.68 | 89.18 |
| MSE | 425.14 | 424.59 | 418.32 | 409.84 | 422.79 | 418.36 | 415.41 | 408.48 | 407.36 | 402.63 | 408.50 | 403.43 | 405.10 | 406.47 | 407.11 | 399.27 |
| NAE | 0.72 | 0.62 | 0.59 | 0.54 | 0.50 | 0.46 | 0.44 | 0.42 | 0.42 | 0.41 | 0.41 | 0.42 | 0.40 | 0.40 | 0.41 | 0.38 |

we implemented a trio of augmentation strategies. The first strategy, Random Augmentation (RA), involves stochastic interpolation of training images. This method introduces a degree of randomness to the scaling of images, which is intended to simulate the diversity of scales that a model might encounter in real-world scenarios. By training the model on this augmented dataset, we aimed to promote the development of scale-invariant features within the model's architecture.

Our second strategy, Inference Augmentation (IA), diverges from the training phase and is applied directly during inference. In this approach, test images are modified with resolution changes akin to adversarial perturbations. This is intended to test the robustness of the model against unexpected scale shifts at inference time, simulating a form of stress test for the model's generalizability.

Our findings imply that image interpolation, while beneficial, should be considered as one component in a multifaceted approach to enhancing scale invariance in image recognition. Further research is needed to explore combinations of interpolation with other techniques, such as multi-scale architectures or hybrid training protocols, to develop more robust solutions capable of handling the diverse scaling challenges present in real-world image datasets. By providing this more detailed explanation of our methodology and results, we hope to convey the nuances of our study's contributions to the field of image recognition and the ongoing efforts to overcome the hurdles of scale variability.

# G  DISCUSSION FOR DIFFERENT KINDS OF DOMAIN SHIFT

In this section, we collect datasets from JHU (Sindagi et al., 2020), and split it into several datasets according to the domain shift type, including scene shift (from Stadium to Street), weather shift (from Sunny to Snowy), dataset shift (from SHHA to QNRF), and count shift (from Dense to Sparse). According to the results that generalizes *source 1* to *target*, we observe non-trivial performance degradation. In the meanwhile, we illustrate the scale distribution divergence between *source 1* and *target*. We notice a correlation between scale divergence with performance degradation. To further support this empirical observation, we manually manipulate the source domain scale distribution to make it farther to the target domain and form a new domain *source 2*. When generalizing from

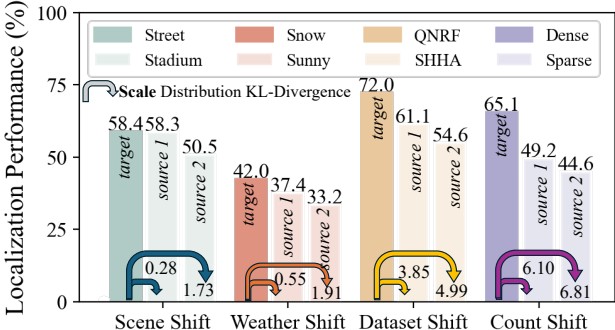

Figure 12: Localization performance of Gao et al. (2020) on the test set of the target domain. Under the same kind of shift, different color depths represent various training sets. The key difference between *source 1* and *source 2* is that we manually replaced certain images in *source 1* to create *source 2*, which features a larger scale shift to the target domain.

*source 2*, we notice a consistent per-
formance degradation. This reveal a significant empirical conclusion: **Scale shift generally exists and coupled with other domain shifts.** This further strengthen the significance in researching this issue.

# H   DISCUSSION ON ANNOTATED DATA

## H.1   DATA ANNOTATION

### H.1.1   ANNOTATION TEAM

To manually annotate over 2,700 images from the SHHA, SHHB, and QNRF datasets, we assembled a team of 39 annotators. All annotators hold at least a bachelor's degree or are undergraduate students, ensuring a level of educational background that we believe is essential for maintaining high annotation quality.

### H.1.2   ANNOTATION PLATFORM

Thanks to the authors of NWPU (Wang et al., 2020b), who have open-sourced a Python Django-based framework for crowd image annotation, we had a convenient platform for this process.

### H.1.3   ANNOTATION PROCESS

Recognizing that annotating bounding boxes in congested and complex scenes can be tedious, we conducted four rounds of annotation. The first round involved initializing bounding boxes based on the method presented in (Gao et al., 2020). The second round focused on refining these boxes through human input. The final two rounds aimed at further refining the manually annotated boxes.

To facilitate this process, we divided our team into three sub-teams: Team A (20 members), Team B (15 members), and Team C (4 members). Through this collaborative approach, we successfully provided more than 1.5 million bounding boxes for the over 2,700 images.

## H.2   UNIFIED EVALUATION METRIC FOR CROWD LOCALIZATION

In the domain of crowd localization, accurately evaluating performance is crucial. Typically, this evaluation involves establishing a point-to-point correspondence between the predicted coordinates and the actual ground-truth positions through the construction of a bipartite graph. Subsequently, distances are computed between paired points, and a prediction is deemed correct if this distance falls below a predetermined threshold. Nonetheless, the choice of threshold is pivotal, greatly impacting the perceived precision of predictions. A threshold that is excessively lenient may yield results that are overly generalized, while an overly strict threshold might result in an underestimation of the model's predictive capabilities. In practice, for datasets that provide bounding box annotations, such as NWPU-Crowd (Wang et al., 2020b) and JHU-Crowd++ (Sindagi et al., 2020), the threshold is often pragmatically set to the length of the diagonal of these boxes. However, earlier datasets like SHHA (Zhang et al., 2016), SHHB (Zhang et al., 2016), and QNRF (Idrees et al., 2018) do not offer such annotations, thereby introducing an element of subjectivity into the evaluation process concerning the localization threshold. In our work, we address this inconsistency by contributing bounding box annotations for the SHHA, SHHB, and QNRF datasets. This enhancement enables us to standardize the evaluation procedure by setting the matching threshold to the diagonal length of the bounding box. Furthermore, we advocate for subsequent research in this area to utilize these annotations, fostering a more objective and uniform assessment of methodological performance across the SHHA, SHHB, and QNRF datasets.

## H.3   TYPICAL SAMPLES FROM ANNOTATED DATASETS

Please see supplementary materials for high resolution images.

