# OpenReview forum: "Understanding Scale Shift in Domain Generalization for Crowd Localization"
_ICLR.cc/2025/Conference — Submitted to ICLR 2025_

### Official Review · Reviewer_Hohg · 2024-11-02

**Soundness:** 2
**Presentation:** 3
**Contribution:** 2
**Rating:** 5
**Confidence:** 4

**Summary:**

This paper analyzes domain generalization under scale shift in crowd localization, where object scales vary across domains. To address the lack of benchmarks for studying scale-related shifts, the authors introduce Scale Bench. This benchmark divides data into domains based on scale and evaluates models on their ability to generalize to unseen scales. They propose Semantic Hook, a training method that uses noise perturbations to reduce scale reliance and strengthen semantic associations in model predictions.

**Strengths:**

1- The paper is well-written and engaging, and the ideas flow smoothly.

2-The field of crowd counting/localization would benefit from an analytical work focused on the issue of scale variance, as scale shifts present a significant challenge for model generalization across diverse domains. This paper addresses this gap, and provides both a theoretical framework and a practical benchmark.

**Weaknesses:**

1- There is limited mention (with brief explanation for each) of papers explicitly in crowd counting/localization fields that tackle the issue of scale variance.

2- While the paper provides a comprehensive benchmark for scale-related domain generalization, it lacks coverage of crowd counting/localization methods specifically designed for domain generalization. How many of the methods in Table 2 discuss scale variance for crowd counting specifically?

3- The paper does not clarify what has been done to prevent overfitting, particularly given the possible complexity of the models relative to the training data provided.

4- Although Tables 6-18 and a brief discussion for each are included in the appendix, the paper lacks an in-depth analysis explaining why certain methods outperform others in specific cases. A discussion of these results would add valuable context to understand the strengths and limitations of each approach under different scale conditions. what could be the issue that each of these methods fail in generalizing to new domain?

**Questions:**

1-What is the difference between semantic hook and other methods that also purturb the image for crowd counting/localization?

2- What other methods specifically in crowd counting and/or localization exist that have addressed the scale variance? Have any of these methods been implemetned in this paper?

3- Why does each of the previous methods fail in addressing this issue? What is the authors insight in this matter?

4- How were the hyperparameters for each model set? Did the authors use grid search?

---

### Official Review · Reviewer_gsTA · 2024-11-03

**Soundness:** 2
**Presentation:** 3
**Contribution:** 2
**Rating:** 5
**Confidence:** 4

**Summary:**

The paper focuses on the impact of scale variations on crowd localization models' ability to generalize across datasets. To tackle this, the authors introduce ScaleBench, a new benchmark dataset specifically curated to study scale shift, and evaluate 20 existing domain generalization algorithms, showing that many struggle with this type of shift. They also propose an approach, Semantic Hook, aimed at mitigating scale shift by strengthening the association between semantic features and predictions, rather than relying on scale information. While the improvements are modest, the paper offers valuable insights into scale-based generalization challenges.

**Strengths:**

1) The development of ScaleBench is a major contribution, offering a curated dataset specifically designed to study scale shift effects on domain generalization.
2) The paper introduces the Semantic Hook as a novel approach to reduce the impact of scale shift in domain generalization tasks.
3) The paper is well-structured and logically organized. Offering theoretical insights and comprehensive empirical evaluations.

**Weaknesses:**

1) The paper frames the issue of scale shift as a new challenge within domain generalization for crowd localization, but this framing seems overstated. Scale shift, where different head sizes (scales) impact model performance across datasets, is not fundamentally new. Previous works have already explored the impact of scale variation on domain adaptation in crowd analysis (albeit under different terminologies), suggesting that this issue is more a subset of a well-studied generalization problem rather than a novel concept. Claiming it as the "first study" on "scale shift domain generalization" could be seen as an attempt to rebrand existing challenges without sufficient justification.

2) The proposed "Semantic Hook" technique to mitigate scale shift claims to enhance the association between semantic features and task predictions, but its practical effectiveness remains questionable. This method involves adding Gaussian noise to "hook" relevant features, yet the theoretical rationale behind this approach is underdeveloped. How "Semantic Hook" contributes to decoupling scale-related biases from semantic content is unclear. Additionally, the improvement in F1 scores presented in Table 2 is marginal, suggesting that the Semantic Hook might not be a robust solution.

3) While the paper provides a comparison of 20 domain generalization algorithms, there is little discussion about the practical differences in their robustness against scale shifts. The Semantic Hook’s performance is only marginally better than ERM, raising doubts about its practical value. Furthermore, the experiments rely heavily on F1 scores across ScaleBench domains but do not include additional evaluation metrics (e.g., precision, recall) that could provide a fuller picture of model performance under scale shift.

4) ScaleBench, with its scale-based domain partitions, may not accurately reflect real-world applications where scale distributions are more complex and continuous rather than discretely defined. The Leave-One-Out approach used for evaluation also artificially simplifies the generalization challenge. Real-world scenarios often involve more nuanced and diverse shifts between training and deployment environments, suggesting that the paper’s evaluation may lack external validity.

**Questions:**

1) How does "scale shift" in crowd localization fundamentally differ from other types of domain shifts?
2) How does Semantic Hook compare with simpler baseline methods, such as multi-scale training or augmentations?
3) Can the spurious association between scale and the output be quantified?
4) How would ScaleBench and Semantic Hook perform in real-world crowd localization scenarios with continuous scale distributions?
5) What are the limitations of ScaleBench in generalizing to diverse crowd analysis tasks?

---

### Official Review · Reviewer_p2oH · 2024-11-04

**Soundness:** 2
**Presentation:** 3
**Contribution:** 3
**Rating:** 5
**Confidence:** 4

**Summary:**

This paper aims to study the effect of scale shift in crowd localization. To this end, a benchmark, dubbed ScaleBench, is first established to quantify the influence of scale shift. Next, SemanticHook is proposed to tackle scale shift. The key idea is to enhance the association between semantic features and targets by perturbing the input image with Gaussian noise. Empirical analyses on ScaleBench justify the effect of scale shift.

**Strengths:**

1. A controlled benchmark is established to study scale variance in crowd localization.
2. This paper proposes SemanticHook to handle scale shift.
3. Comprehensive analyses are presented to quantify the influence of scale shift.

**Weaknesses:**

1. The rationality of ScaleBench is questionable. First, while the perspective effect often occurs in crowd localization, there exist images that are captured from different angles (e.g., top view). In such scenarios, image distribution regularization may fail to partition the images correctly. Second, in the real world, scale shift is often coupled with other factors, such as occlusion, weather, and appearance. For example, when the object suffers from significant appearance variations, the counting model may fail to localize objects even if training and testing data yield the same scale distribution. Third, dividing images into patches will inevitably result in incomplete objects, which could affect the localization results. Therefore, evaluations on ScaleBench may not rigorously reflect the influence of scale shift.
2. The proposed SemanticHook does not exhibit superiority over existing methods. As shown in Table 1, the simplest baseline ERM already achieves good results. The proposed method is not necessarily better than ERM.
3. Following the previous comment, the rationale of SemanticHook is not entirely convincing. Eq. 6 suggests that p(s, c, …) can lead to a spurious association between the output y and scale c. This term is a joint distribution of semantic s and scale c. However, the authors merely try to enhance the semantic association between semantic s and output y. Experimental results demonstrate that such a technique does not address scale shift effectively. Additionally, perturbing image is not a new idea, which is widely used in adversarial attack.
4. It appears that the influence of image interpolation is not rigorously quantified in Table 4. First, the implementation of Random Augmentation shall be modified according to different domains, i.e., the range of random scaling should be customized based on domain Tiny, Small, and Normal. Second, it is necessary to train the model using different source domains to identify the effect of image interpolation. The results on domain Big are insufficient to conclude that the benefits of image interpolation are modest.
5. Regarding training details. In practice, random scaling is commonly used to alleviate scale variations. As the authors use this technique to train the model, the reported results may not correctly reveal the effect of scale shift, because random scaling already simulates different scales.
6. The paper lacks evaluations on previous methods featuring multi-scale architecture, e.g., STEER. Evaluations on these methods are helpful in revealing whether previous methods can handle scale variations.

**Questions:**

1. What does global feature mean in Table 5? Figure 2 shows that semantic features are extracted from the encoder. How to extract global feature?
2. Is the proposed method sensitive to the choice of gamma?

---

### Official Review · Reviewer_RxyN · 2024-11-04

**Soundness:** 3
**Presentation:** 3
**Contribution:** 2
**Rating:** 3
**Confidence:** 5

**Summary:**

The paper investigates the impact of scale shift on domain generalization in crowd localization. Models experience performance degradation due to head scale distribution shifts in training and testing datasets. To address this, the authors provide a theoretical analysis of the scale shift under domain generalization and introduce a novel method to mitigate the effect of scale shift, called Semantic Hook. The paper proposes a new benchmark called ScaleBench and provides bounding box annotations for existing public crowd benchmarks.

**Strengths:**

The paper addresses an under-explored issue of scale shift in domain generalization for crowd localization. In terms of contributions, the paper delivers manually annotated bounding boxes for crowd localization on existing public crowd benchmarks. The paper is well-structured and provides a good analysis of the problem, resulting in a novel solution method called Semantic Hook. Further, the authors take an analytical route for the scale shift under domain generalization connecting other attributes present in datasets.

**Weaknesses:**

The paper needs more detailed explanations regarding how Semantic Hook mitigates the scale shift in domain generalization. It also needs to clarify which variables or attributes are being generalized from the perturbation added during training. Additionally, the improvements from the proposed method on the ScaleBench benchmark are marginal compared to the baseline method. Furthermore, the mathematical formulations (Eq. 6) used for the theoretical analysis need to be corrected.

**Questions:**

1. Explain how Semantic Hook handles the scale shift on domain generalization. From the given formulation, the semantic difference will highlight the effect of the perturbation, and the decoder is now learning to map noise to task-specific outcomes. So, how does Semantic Hook reduce generalization risk?
2. The conditional probability derived in Eq. 6 is incorrect at the first integral. The conditional probability P(y|x) does not equal integrating P(y|z) over the domain of Z. Please provide the correct derivation.
3. The scale shift is more prevalent in crowd images under perspective projection; however, the scale is more uniform throughout the scene for aerial views. How does the proposed method handle different projections?

---

### Official Review · Reviewer_pL1Y · 2024-11-04

**Soundness:** 3
**Presentation:** 3
**Contribution:** 3
**Rating:** 5
**Confidence:** 4

**Summary:**

This paper presents a significant contribution to domain generalization (DG) for crowd localization by addressing the challenge of scale shift, where differences in head size distributions between training and testing data impact model performance. The authors introduce ScaleBench which categorizes datasets based on scale distributions. They also propose Semantic Hook, an algorithm designed to mitigate scale shift by reinforcing the association between semantic features and task predictions. Through testing 20 state-of-the-art DG algorithms on ScaleBench and conducting theoretical analysis, the authors highlight the limitations of current approaches and introduce Scale Shift Domain Generalization as a novel research direction.

**Strengths:**

1. The identification of scale shift as a specific domain shift challenge in crowd localization, and the introduction of Scale Shift Domain Generalization, bring attention to an under-explored issue with significant real-world implications. ScaleBench provides a standardized benchmark, adding practical value for the research community.
2. The authors provide a clear theoretical explanation linking scale shift to diversity and correlation shifts, elucidating why DG models struggle with this issue. This rigorous analysis adds depth to the understanding of scale shift and its implications for DG.
3. The paper is well-organized, with each section following logically from the last. The clear delineation between problem identification, analysis, and solution makes the contributions easy to follow.

**Weaknesses:**

1. The introduction lacks highlighting core contributions and findings.
2. Although the authors indicate that the paper does not primarily focus on introducing a new method, the experiments in the main text feel somewhat limited.
3. The appendix contains several formatting issues, particularly with tables. Inconsistencies include varying font sizes, tables floating in the middle of pages, and some tables exceeding the page width. These layout problems affect readability and detract from the paper's presentation quality.

**Questions:**

I have no further questions beyond those outlined in the weaknesses section.

---

> ### Comment · Reviewer_pL1Y · 2024-12-02
>
> Since the authors did not engage in the rebuttal process, I am inclined to maintain a somewhat negative rating.

---

### Meta-Review · Area_Chair_afKu · 2024-12-19

**Metareview:**

This paper was reviewed by five experts in the field. The final ratings are 3,5,5,5,5. Reviewers generally agree that the scale shift is an important and interesting problem in crowd localization. Reviewers also raised many concerns, including insufficient explanation of the proposed method, limited experiments, etc. There is no rebuttal from the authors, so there is no ground to overrule reviewers' recommendations.

**Additional Comments On Reviewer Discussion:**

The authors did not provide rebuttal

---

### Decision · Program_Chairs · 2025-01-22

Reject